# Implementation of the MOSAIC Aerosol Module (v1.0) in the Canadian Air Quality Model GEM-MACH (v3.1)

Kirill Semeniuk[1], Ashu Dastoor[1], and Alex Lupu[1]

[1]Air Quality Research Division, Modeling and Integration Section, Environment and Climate Change Canada

*Correspondence to*: Kirill Semeniuk (kirill.semeniuk@ec.gc.ca)

**Abstract.** The Model for Simulating Aerosol Interactions and Chemistry (MOSAIC) aerosol thermodynamics and sectional framework has been implemented into the Canadian operational air quality model GEM-MACH.    The original aerosol sub-model in GEM-MACH is based on the Canadian Aerosol Module (CAM), which uses a single-

moment (mass) sectional scheme, and inorganic thermodynamics derived from the equilibrium ISORROPIA model without base metal cations.  MOSAIC features non-equilibrium inorganic thermodynamics and a double-moment (mass and number) sectional scheme.   For evaluation we conduct four one-year simulations with the same emissions and meteorology over the North America domain.   A reference run (REF) with the Zhang et al. (2001) aerosol dry deposition scheme and a sensitivity run (EMR) with updated parameters from Emerson et al. (2020) is

conducted for each aerosol model option.     The results are compared to station observations and surface monthly-mean model-observation synthesis data.   MOSAIC exhibits a shift in the accumulation mode mass and number distribution compared to CAM that results in more aerosol dry deposition in the REF run and a surface PM2.5 sulfate low bias of about 15% relative to CAM.  This bias is essentially removed in the MOSAIC EMR run resulting in a better fit to aggregated urban and rural stations compared to CAM over the North America domain.

Comparison with the AERONET volume size distribution inversion product shows that MOSAIC gives a much higher level of agreement in terms of location of the accumulation mode peak diameter and separation of the accumulation and coarse modes.   PM2.5 nitrate and ammonium for the MOSAIC EMR run shows overall better agreement with observation station data compared to both REF and EMR CAM runs at rural stations.   At urban stations MOSAIC has a high bias for nitrate relative to CAM and observations during summer but it is reduced in

the EMR run compared to the REF run.   The high bias in ammonium seen with CAM for both REF and EMR runs relative to aggregated rural and urban station observations is reduced with MOSAIC by about 25% between April and November.

## 1 Introduction

Ongoing advances in physical process understanding spur revision of air quality models and other chemistry-transport models with improved parameterizations.  This improves representation of atmospheric composition and enables more realistic process coupling (e.g. Baklanov and Zhang, 2020; Shrivastava et al., 2017).   Improvements in computational resources help offset the increased numerical cost of more detailed parameterizations (e.g. Nakaegawa, 2022).     Here we describe work to update the inorganic aerosol scheme in the Environment and

Climate Change Canada (ECCC) air quality model GEM-MACH (Global Environmental Multiscale – Modeling Air Quality and Chemistry). This is part of a project that aims to enable interactive meteorology-chemistry modeling, which also includes upgrades to the nucleation (inorganic and organic) and organic thermodynamics process representation. The revised model is intended for policy scenario and research simulations instead of operational forecasting due to the high numerical cost of these process representation improvements.

The GEM-MACH aerosol sub-model consists of the single-moment CAM scheme (Gong et al., 2003) and the inorganic thermodynamics scheme HETV (Makar et al., 2003) derived from ISORROPIA I (Nenes et al., 1998). A single-moment mass-only formulation limits the capacity for process coupling such as cloud-aerosol interaction which requires a more accurate representation of aerosol number (e.g. Li et al., 2008). HETV does not consider the Kelvin curvature effect which impacts mass transfer for sub 100 nm diameter aerosols in the nucleation and Aitken modes. Mass fluxes over the fine particle sizes can be comparable to or exceed mass fluxes over the accumulation and coarse modes (Zaveri et al., 2008). In GEM-MACH, HETV is applied to the bulk aerosol mass instead of individual size bins to save computational expense. The bulk result is unpacked into bins using gas ($H_2SO_4$, $HNO_3$ and $NH_3$) transfer rate fractions (Fuchs and Sutugin, 1971) to individual size bins. For each constituent gas species, the fractions are based on bin size transfer rates (e.g. Equation 5 in Zaveri et al., 2008) divided by the sum of these rates over all bin sizes.

To address limitations in mass transfer and lack of prognostic aerosol number, we have implemented MOSAIC (Zaveri et al., 2008). A comprehensive comparison between MOSAIC and ISORROPIA thermodynamics is given in Zaveri et al. (2008). The components required for a double-moment scheme are the coagulation and sectional adjustment routines, which need to track aerosol number. For the internally mixed aerosol formulation, aerosol number requires a tracer for each size bin. Details of GEM-MACH and revisions are given in Sections 2 and 3. Comparison of the original aerosol scheme (CAM + HETV) and MOSAIC is conducted by way of one-year simulations on the North America regional domain with a typical resolution of 10 x 10 km. The model PM2.5 and total PM particulate output is evaluated using observational surface station network data and regional speciated PM2.5 distribution estimates from a combined geoscience-statistical method (van Donkelaar et al., 2019). The choice of the latter is a way to fill the gap in spatially distributed observations, but it will reflect model biases. Validation against station data is necessarily limited by the fact that the model does not resolve spatial scales associated with station measurements. The range of concentrations seen by the model is smaller than observations as sub-grid plumes are not resolved (e.g. Sun et al., 2021). In addition, emissions are limited in spatial and temporal resolution as well, and minor localized emissions are aggregated into area emissions (e.g. Kuenen et al., 2022). Model results and comparison with observational products are presented in Section 4.

## 2 Description of GEM-MACH

GEM-MACH is an air quality extension of the GEM forecast model (Girard et al., 2014, and references therein). For this work the version of MACH used is 3.1.0a.2 and the version of GEM is 5.1.2. The GEM meteorological

70 physics package (radiative transfer, convection, precipitation, land surface interaction, etc.) is described by Mailhot et al. (1998) and recent improvements by McTaggart-Cowan et al. (2019). In the simulations presented here, shallow convection is parameterized with the scheme from Bechtold et al. (2001) and deep convection by the scheme of Kain and Fritsch (1990).

Currently, the chemical and aerosol tracers are not transported directly by parameterized convection. This is
75 mitigated to a significant degree by vertical transport associated with resolved mesoscale and synoptic systems (e.g. Polvani and Esler, 2007; Lyons et al., 1995). However, lack of tracer convection does introduce a low bias in the transfer of tracer mass into the free troposphere from the atmospheric boundary layer via shallow convection. This aspect is discussed in the study of Polavarapu et al. (2016) which focuses on $CO_2$ transport in a GEM-MACH derived model.

80 The air quality package, MACH, includes modules for gas and aqueous phase chemistry, inorganic aerosols, secondary organic aerosols (SOA), and wet and dry removal of gases and aerosols. Inorganic aerosol processes are represented by CAM and HETV in 2-bin (operational) or 12-bin configurations (scenario and research). SOA formation is based on the instantaneous yield model of Jiang (2003). Following the common approach, aerosols in GEM-MACH are formulated as internal mixtures to reduce computational expense. The gas phase chemistry
85 options are ADOM-II (Venkatram et al., 1988), SAPRC07 (Carter, 2010) and SAPRC11 (Carter and Heo, 2013). For this study the ADOM-II chemistry option is used for both CAM and MOSAIC. The aqueous phase chemistry component (Gong et al., 2006) is derived from ADOM (Karamchandani et al., 1985) and is operated in bulk mode. The wet scavenging of gases and aerosols is described in Gong et al. (2006). Dry deposition of gases is described in detail by Makar et al. (2018) and is based on a modification of the Wesely (1989) scheme. The Zhang et al. (2001)
90 scheme is used for dry deposition of aerosols.

Currently, there is no heterogeneous hydrolysis of $N_2O_5$ into $HNO_3$ in GEM-MACH. This is a significant source of particulate nitrate (Chang et al., 2011; Kim et al., 2014) and will be included in a future model update.

GEM-MACH with the CAM option has been evaluated against observations by Im et al. (2015a, b), Makar et al. (2015), Gong et al. (2015), Whaley et al. (2018), and Majdzadeh et al. (2022). GEM-MACH compares reasonably
95 well to other air quality models and observations.

**3 GEM-MACH Revisions**

**3.1 Aerosol Model**

MOSAIC version 1.0 with updates was used for this work. The original source code was extracted from the
100 chemistry extension of the Weather Research and Forecasting Model (WRF-Chem) version 3.2 (Skamarock et al., 2008). The updates are based on the MOSAIC (v1.0) revision found in WRF-Chem version 4.0.3 (Skamarock et al., 2019) that pertain to bug fixes and the phase state in the Multicomponent Equilibrium Solver for Aerosols (MESA) module (Zaveri et al., 2005). MOSAIC was also modified to include primary emission carbon (PC) as an additional constituent.

MOSAIC includes routines for nucleation, aerosol thermodynamics (including sulfuric acid condensation), coagulation (Jacobson et al., 1994), and sectional adjustment. A comparison of CAM and MOSIAC as used in this study is given in Table 1. SOA formation is treated via optional simple (Hodzic and Jimenez, 2011) and in later versions more complex schemes (Shrivastava et al., 2011). Nucleation options are the Wexler et al. (1994) scheme and a combined scheme employing binary nucleation (Vehkamäki et al., 2002) and ternary nucleation with ammonia

(Merikanto et al., 2007) depending on available ammonia. A post-nucleation growth (PNG) model (Kerminen and Kulmala, 2002) is used to populate the first size bin with nucleated aerosol. Sectional adjustment for bin mass and number from condensation and evaporation in MOSAIC can be selected from the Simmel and Wurzler (2006) mass-number advection approach and the Jacobson (1997) moving-center approach.

**Table 1.** Comparison of CAM and MOSAIC

| Model Feature | CAM | MOSAIC |
|---|---|---|
| Scheme order | Single-moment (mass) | Double-moment (mass and number) |
| Coagulation | Volume-conserving, semi-implicit (Jacobson et al., 1994); Brownian, turbulence and gravitational settling kernels | Volume-conserving, semi-implicit (Jacobson et al., 1994); Brownian kernel. |
| Sectional adjustment | Based on coagulation scheme: condensing species volume and existing bin volume are treated as particle merger. | Moving-center (Jacobson, 1997) |
| Thermodynamic system | $SO_4 + NH_4 + NO_3 + H_2O$ | $SO_4 + NH_4 + NO_3 + Na + Cl + Ca + CO_3 + H_2O$ |
| Irreversible reactions | No | Yes; formation of particle phase $CaCl_2$, $CaSO_4$, $Na_2SO_4$, $Ca(NO_3)_2$, $NaNO_3$ |
| Water scheme | Hanel (1976); accounts for Kelvin curvature effect | Zdanovskii-Stokes-Robinson (Zdanovskii, 1948; Stokes and Robinson, 1966); accounts for Kelvin curvature effect |
| Hydration hysteresis | No; interpolation between deliquescence point and crystal state | Yes; MESA (Zaveri et al., 2005) |
| Thermodynamics solution and mass transfer approach | Equilibrium over 12 hydration-acidity regimes; no Kelvin curvature effect | Time-dependent over 18 hydration-acidity regimes; includes Kelvin curvature effect. |
| Thermodynamics size resolution | Bulk; splitting into bins using Fuchs and Sutugin (1971) based weights | Applied to each size bin |


For the GEM-MACH implementation of MOSAIC, we have retained only the thermodynamics, coagulation and sectional adjustment routines. The existing Odum et al. (1996) type SOA scheme (Jiang, 2003) is used with MOSAIC instead. Sulfate nucleation and condensation is based on the scheme from CAM and is described in more detail in the next subsection. These choices serve to reduce model differences.

MOSAIC calls thermodynamics for every sectional bin and we chose not to apply the bulk approach as used for HETV in CAM. A bulk approach would substantially reduce numerical expense, but it defeats the purpose of accounting for the Kelvin curvature effect and degrades the accuracy of the model. This is addressed in Section 4.3.

The coagulation scheme in CAM follows the volume-conserving, semi-implicit formulation of Jacobson et al. (1994) which is used by MOSAIC. However, the formulation in CAM is single moment and aerosol number is not

conserved. This applies to the sectional adjustment as well (Gong et al., 2003). For the internally mixed formulation, CAM aerosol number for each size bin is determined from the total dry aerosol volume in the bin and the bin volume based on the average bin radius. MOSAIC assumes internal mixing as well but requires the addition of a number tracer for each size bin. The MOSAIC coagulation and sectional adjustment components conserve aerosol number.

For this study, MOSAIC is used with the moving-center sectional adjustment option (Jacobson, 1997). This is a less numerically expensive scheme compared to mass-number advection. The moving-center scheme has been found to suffer from gap formation between bins depending on bin resolution, including the 12 bins used here (Mohs and Bowman, 2011). However, we did not find any pathology in our simulations. We conducted box model tests with different adjustment schemes (not shown) with realistic initial conditions and sulfate production and did not

find a gap formation issue with the moving-center scheme. Thus, this pathology is not a generic feature and requires specific conditions to occur and this is reflected in our GEM-MACH results. With both sectional adjustment options, we needed to introduce double precision to the MOSAIC code to avoid issues with grid level noise. The GEM-MACH source code is compiled in single precision which requires precision sensitive routines to have explicit double precision coding.


### 3.2 Sulfate Nucleation and Condensation

For the MOSAIC option we have adapted the sulfate nucleation and $H_2SO_4$ condensation scheme used in CAM. The nucleation from sulfuric acid is calculated simultaneously with condensation onto existing aerosol to better capture the competition between nucleation and condensation scavenging. This involves solving approximations of

the condensation and nucleation equations simultaneously by time-stepping over 15 variable time sub-intervals over every GEM-MACH chemistry time step as outlined in Gong et al. (2003).

The binary sulfate nucleation scheme from Kulmala et al. (1998) is used for both CAM and MOSAIC. There is no PNG parameterization in CAM and nucleated sulfate mass is introduced into the first model bin. For simplicity we have not included the PNG scheme from MOSAIC in the present study. For MOSAIC the nucleation aerosol

number is calculated based on nucleated sulfate volume in the first size bin and the bin volume derived from the lower limit of the bin radius, which is 5 nm. Work to overhaul the GEM-MACH aerosol nucleation to reflect recent advances and to include a post-nucleation growth scheme will be described elsewhere.

### 3.3 Aerosol Constituents and Bin Size Distribution

CAM has eight aerosol constituents: sulfate (SU), ammonium (AM), nitrate (NI), crustal material or soil dust (CM), sea salt (SS), primary emitted organic carbon (PC), secondary organic carbon (OC) and elemental or black carbon (EC). Aerosol water and aerosol number are diagnostic fields in CAM. CAM aerosol number is derived from aerosol dry volume and bin size. The aerosol water used for coagulation and scavenging processes is not taken from HETV but calculated via the scheme described in Appendix A of Gong et al. (2003) which is based on Hänel

(1976). This scheme takes into account the Kelvin curvature effect for small diameter particles. To address the lack of particle RH exposure history the hydration growth factor is linearly interpolated between the deliquescence and crystallization points (Gong et al., 2003).

MOSAIC has an expanded list of aerosol constituents compared to CAM but lacks primary emitted organic carbon. As noted in section 3.1, we extend MOSAIC to include PC. The MOSAIC BC (black carbon) and OC (organic

carbon) constituents are mapped into the EC and OC constituents as used by CAM. Crustal material is split into calcium (CA), carbonate (CB) and other inorganic matter (OI). Sea salt is represented by sodium (NA) and chloride (CL). Methane-sulfonic acid (MSA), aerosol number (NU), aerosol water (WA) and aerosol hysteresis water (HW) are introduced as transported constituents. We do not have emissions and chemistry for MSA in GEM-MACH, so for this study MSA is not used. Aerosol water uptake in MOSAIC uses the Zdanovskii-Stokes-Robinson (ZSR)

method (Zdanovskii, 1948; Stokes and Robinson, 1966) and accounts for the Kelvin curvature effect in the calculation of the water activity (Zaveri et al., 2008). In the MOSAIC version used here we do not include the contribution of organics to water uptake for consistency with CAM. In contrast to the CAM option, MOSAIC aerosol water is subject to hysteresis effects as determined by the comprehensive MESA sub-model (Zaveri et al., 2005).

The same 12-bin size distribution is adopted for both CAM and MOSAIC, with bin limits specified in radius (microns) by: $0.005 \cdot 2^{(n-1)}$, n = 1-13. Each constituent is represented by 12 tracers totaling 96 for CAM and 168 for MOSAIC.

### 3.4 Surface Emissions

The gas and bin-resolved aerosol emissions processed for the Air Quality Model Evaluation International Initiative (AQMEII) Phase 4 project (Galmarini et al., 2021) are used for both the CAM and MOSAIC options. The surface emissions for gases and aerosols consist of area and major point sources (stack emissions). Stack emissions are distributed into the near surface domain using a plume-rise model (Akingunola et al., 2018). Area emissions are distributed using vertical diffusion. Fire emissions are handled as major point sources and subjected to the Briggs

plume-rise scheme from the Community Multiscale Air Quality (CMAQ) model (Li et al., 2023). Biogenic area emissions of plant non-methane volatile compounds (VOC) and soil $NO_x$ are calculated online. Vegetation distributions are from the Gridded Biogenic Emission Land Use Database (BELD) version 3 (Pierce et al., 2000).

Emission rates from the Biogenic Emissions Inventory System (BEIS) version 3.09 (Vukovich and Pierce, 2002) are normalized for use with model predicted meteorological conditions.

Non-point emissions used in this study include anthropogenic fugitive dust with meteorological modulation and road emissions with the effect of vehicle induced turbulence accounted for (Makar et al., 2021). Meteorological modulation is parameterized as a weighting factor which is zero if the grid cell ground moisture fraction is over 10% and has a value between 0 and 1 depending on the grid snow cover fraction. Only anthropogenic area and major point source dust emissions are accounted for. GEM-MACH currently lacks an online dust emission scheme for

non-anthropogenic sources. Sea salt emissions are generated online using the Gong-Monahan scheme (Gong et al., 2002).

Dust emissions are speciated and consist of Ca, Mg, K, Na, Fe, Mn and the remainder (CM). They are used by the aqueous chemistry scheme for all simulations. However, the HETV version used with CAM does not use these metal cations and they are lumped into CM. For MOSAIC, soil Na, Mg, and K are lumped into NA, but Fe and Mn

are lumped into OI following the prescription in Zaveri et al. (2008). Lumping is done on a molar basis and conversion to mass assumes the molar mass of the target tracer. The carbonate emissions for MOSAIC are inferred from Ca assuming that it is part of $CaCO_3$. In reality, $CO_3$ is not restricted in this fashion and can exist in association with other base cations (e.g. Doner and Lynn, 1989) but we do not have the detailed crustal material characterization available for this study. There are dust and fire emissions of chloride, but these are not accounted

for in our emissions inputs.

If the cation species were included as individual size bin tracers, then for CAM this would add 72 tracers and 48 for MOSAIC (since Na and Ca are already included). To reduce the computational cost, bulk tracers are introduced for each of the six cation emissions. For MOSAIC a bulk tracer for Ca is not included but bulk crustal Na is added to keep it distinct from sea salt sodium for use with the aqueous chemistry. The CM (CAM) or OI (MOSAIC) bin

mass distribution is used to split the bulk tracers into bins. The emission, transport and loss of the metal fractions follows that of these aggregate constituents and there are no additional model processes which modify the bin distribution in a cation-specific way.

Emissions inputs for point and area sources are reprocessed online for MOSAIC. OI is determined from CM by removing the carbonate mass inferred from CA. Sea salt emissions are split into Na and Cl based on the molar

mass. Aerosol number emissions are calculated based on the bin total dry volume (the sum of the dry mass divided by density of each constituent) of emitted species and average bin volume. This is a simplistic approach (e.g. Xausa et al., 2018) but aerosol number emissions are typically not measured and would not conform to the internal mixture assumption of the model. We do not add aerosol water emissions.

**3.5 Chemical Constituent Lateral Boundary Conditions**

Lateral boundary conditions for gases and aerosols produced for the AQMEII-4 simulations (Galmarini et al., 2021) are used for both CAM and MOSAIC. They are taken from the Copernicus Atmosphere Monitoring Service

(CAMS) reanalysis (Inness et al., 2019) and have a three hour temporal resolution. Missing gas and aerosol species from the CAMS reanalysis product were supplied by seasonal means from a MOZART-4 (Emmons et al., 2010) simulation for 2009 with meteorological inputs from GEOS-5 (Molod et al., 2015). MOZART is a chemistry transport model using different chemistry and physics routines than those in GEM-MACH. In the future, global GEM-MACH simulations will be used to produce high temporal and spatial resolution boundary conditions.

CAMS and MOZART only have bin-distributed sea salt and dust but other constituents such as sulfate are bulk fields. CAMS provides 3-bin data that spans diameters from 0.06 µm to 40 µm. MOZART provides 4-bin data that spans diameters from 0.1 µm to 10 µm (dust) and 20 µm (sea salt). This low bin resolution data was reprocessed into 12 bins via linear interpolation based on dry-adjusted radius fractions for CAMS. Linear interpolation was also carried out for MOZART data but using log-normal radius distributions to compensate for lack of size range overlap. Bulk constituents were size-distributed as single modes using log-normal functions (Emmons et al., 2010).

To adapt lateral boundary conditions for MOSAIC, the CM and SS constituents were split as follows. The CM field was decomposed into CA, CB and OI assuming calcium is 1:1 associated with carbonate. The $CaCO_3$ fraction of CM is taken as a uniform 2.5% which is reasonable for North America (e.g. Reff et al., 2009). The mass fractions of CA and CB out of this 2.5% of the total mass were determined using the molecular weights of Ca and $CO_3$. SS was decomposed into sodium (NA) and chloride (CL) based on their molecular weight fractions assuming that no other sea-salt constituents are present. Aerosol number was obtained from the total dry aerosol volume as for the emissions.

**3.6 Aqueous Chemistry**

The aqueous chemistry for all aerosol sub-model options makes use of speciated dust emissions. This includes Ca, Mg, K and Na to account for their impact on pH, and Fe and Mn for $HSO_3$ oxidation to sulfate (Ibusuki and Takeuchi, 1987). Base cations affect the sulfate formation rate through the high sensitivity of the $O_3$ oxidation pathway to pH (e.g. Turnock et al., 2019). The scheme does not consider chloride. We have chosen to exclude sea-salt $Na^+$ from MOSAIC in the aqueous chemistry scheme to be consistent with the CAM option and only include the dust emission source as in CAM.

**3.7 Wet and Dry Scavenging of Aerosols and Gases**

For dry deposition of gases, GEM-MACH uses a Wesely (1989) type scheme (see supplement of Makar et al., 2018). The dry deposition of aerosols is handled by the scheme of Zhang et al. (2001). The existing model code was modified to include aerosol number as a deposited species.

For this work we have also introduced the Emerson et al. (2020) (EMR) dry deposition parameters as an additional option. This is motivated by the availability of more comprehensive observational data sets used to update the dry deposition model. Over land, the EMR scheme has a substantially reduced deposition velocity for particles with

diameters less than 500 nm but substantially increased deposition velocity for diameters over 1000 nm when compared to the Zhang et al. (2001) scheme (see Figure 1 in Emerson et al., 2020). Over water there is a substantial reduction in the deposition velocity for PM2.5. The shift in the scavenging minimum from sizes above 1 µm to

around 0.1 µm in the Emerson scheme results in better agreement with observations, which is supported by other studies (e.g. Pleim et al., 2022; Jiang et al., 2023). As discussed below, this has a significant impact on the MOSAIC results due to differences in the size distribution compared to CAM.

MOSAIC includes HCl production in the aerosol phase and degassing through formation of $NaNO_3$. The ADOM-II chemistry option used for this study does not include HCl, but this species was added as a new gas tracer with dry

deposition and wet scavenging. The dry deposition parameters are similar to those of $HNO_3$. The wet scavenging of HCl was represented in the same manner as aerosols since HCl uptake into droplets is not part of the aqueous chemistry formulation. A more comprehensive treatment of HCl chemistry would be preferable but is beyond the scope of this study.

**4 Simulations and Results**

We conducted one-year model simulations for 2016 for the CAM and MOSAIC options on a 10 by 10 km (772 by 642 grid) regional North American domain (see Fig. 3 in Majdzadeh et al., 2022) with 84 staggered hybrid vertical levels extending to about 60 km. To save on computing resources, chemistry and aerosol processes are calculated over the bottom 52 levels which cover the troposphere and the lowermost stratosphere. Above the moist tropopause

(less than 10 ppmv of water) the model uses a linearized chemistry to forecast ozone (McLinden et al., 2000; de Grandpré et al., 2016). The initial tracer state input at 0 GMT on January 1, 2016 for CAM is taken from AQMEII-4 simulations. For MOSAIC this input is reprocessed to match needed constituent inputs and used to conduct a two-week spin-up run to produce a new initial state. Nevertheless, MOSAIC simulations still have an initial adjustment transient which has a small impact on the results presented here.

Dynamical boundary conditions and initial states are taken from global 2016 GEM analyses with the Global Deterministic Prediction System (GDPS; Buehner et al., 2015) with output on a 1249 by 834 grid with 81 levels. For the runs conducted here, the dynamical state is re-initialized every 24 hours using these analyses with a 3-hour adjustment period.

Four simulations were produced: two reference runs, designated as REF, for CAM and MOSAIC use the Zhang et

al. (2001) aerosol dry deposition scheme and two runs using the same inputs but with the Emerson et al. (2020) dry deposition parameters, designated as EMR. Model aerosol and gas fields are saved every hour at the surface and at all model levels once every 24 hours at 0 GMT. We present diagnostics of surface distributions and observation station comparisons of PM2.5 and total PM aerosol in the following subsections. A comprehensive evaluation of gas phase constituents is beyond the scope of this paper, but we include some analysis with station observations in

Section S2 of the supplementary material.

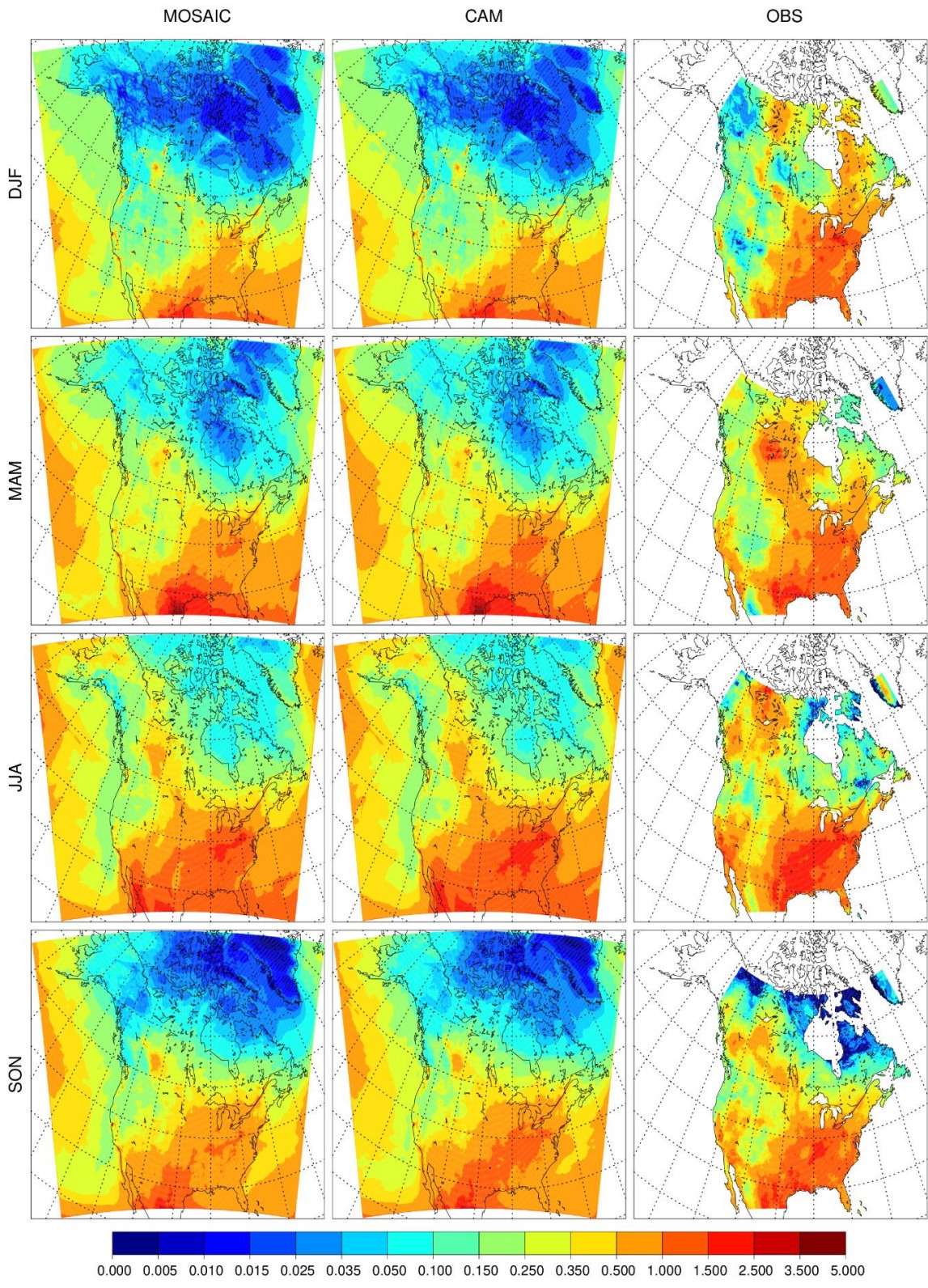

**Figure 1a: Seasonal mean surface sulfate (µg/m³) for the reference MOSAIC run (left), reference CAM run (center) and observation product (right).**

### 4.1 Seasonal-Mean Surface Distributions

In this section we present surface distributions of PM2.5 sulfate, nitrate and ammonium for the REF and EMR model runs and compare them against the model-observation synthesis product from van Donkelaar et al. (2019) (referred to as observation product henceforth). Figure 1a shows the seasonal average surface distribution of sulfate for the reference simulations with MOSAIC (left column), CAM (middle column), and observation product (right column). MOSAIC shows a 10-20% low bias for sulfate compared to CAM over the south-eastern part of North America, especially in summer. Both MOSAIC and CAM have a low bias relative to the observation product in winter (DJF) over most of the domain. There is a substantial low bias over the eastern USA and Canada south of 50°N and a general low bias over the Arctic and sub-Arctic regions around Hudson's Bay. With either aerosol option GEM-MACH, sulfate does not exceed 1 $\mu g/m^3$. But it can exceed 1.5 $\mu g/m^3$ in the observation product. In spring (MAM), the model shows excessive sulfate at the southern boundary of the domain over eastern Mexico but continues to have a low bias over the eastern USA albeit reduced in magnitude. There is also a significant low bias over California. The agreement with the observation product is better in summer (JJA) although there is a low bias to the north of the Gulf of Mexico and over California. In the fall (SON), the model retains the low bias over California and has a low bias to the south of the Great Lakes region.

The summer low bias over the Northwest Territories, British Columbia and northern Alberta and Saskatchewan appears to be at least partly linked to inadequate forest fire emissions and distribution. The treatment of fire emissions for the AQMEII-4 project did not include realistic handling of pyro-convection and fire plumes in general. This is reflected in simulations presented here. Differences between the MOSAIC and CAM sulfate distribution are independent of the thermodynamics scheme since sulfuric acid is a low volatility vapor which is condensed using the same formulation in both cases. As described in Section 4.3, MOSAIC and CAM have substantially different particle size distributions which affect size-dependent removal processes. This becomes readily apparent with the EMR runs (see below).

Figure 1b shows the seasonal nitrate distribution. Due to additional nitrate formation pathways, MOSAIC produces substantially more nitrate over the oceans. MOSAIC includes Na and Ca which buffer the pH of aerosol particles which allows more $HNO_3$ uptake (Karydis et al., 2021). Sea salt chloride is lost via HCl formation which results in a net pH increase. MOSAIC underestimates $NO_3$ in the Great Lakes region in winter compared to CAM and CAM has a low bias relative to the observation product. Observed nitrate exceeds 2.5 $\mu g/m^3$ in this region and 3.5 $\mu g/m^3$ to the south-west of Lake Michigan. The model does not exceed 2.5 $\mu g/m^3$ for either aerosol option. CAM has an overall better agreement with the observation product in the spring but has a high bias to the south of the Great Lakes region in summer. MOSAIC is closer to the observation product in summer. CAM has relatively less bias compared to the observation product in the fall compared to MOSAIC. MOSAIC has excessive nitrate formation over the oceans as can be inferred from the high bias over Baja California and Florida. This appears to reflect the excessive sea salt emissions in the model (Spada et al., 2013; Jaeglé et al., 2011) but dry deposition has an impact as will be discussed below.

The seasonal distribution of $NH_4$ is shown in Figure 1c. The spatial integral of MOSAIC values is substantially smaller than CAM for every season even though the range of concentration values is the same. In winter both

aerosol options produce too little ammonium in the southern Great Lakes region where concentrations can exceed 1 µg/m$^3$. Both CAM and MOSAIC produce too much $NH_4$ in both solstice seasons in the region of Florida and western Mexico. Both aerosol schemes produce too much $NH_4$ in the eastern USA and Great Lakes region in spring, summer and fall. Model values are above the level of 0.5 µg/m$^3$ which is not exceeded in the the observation product. During these seasons the model fails to spread ammonium over California outside of the Los

Angeles region and the Central Valley. This points to model limitations in the transport and mixing in the boundary layer and by convection.

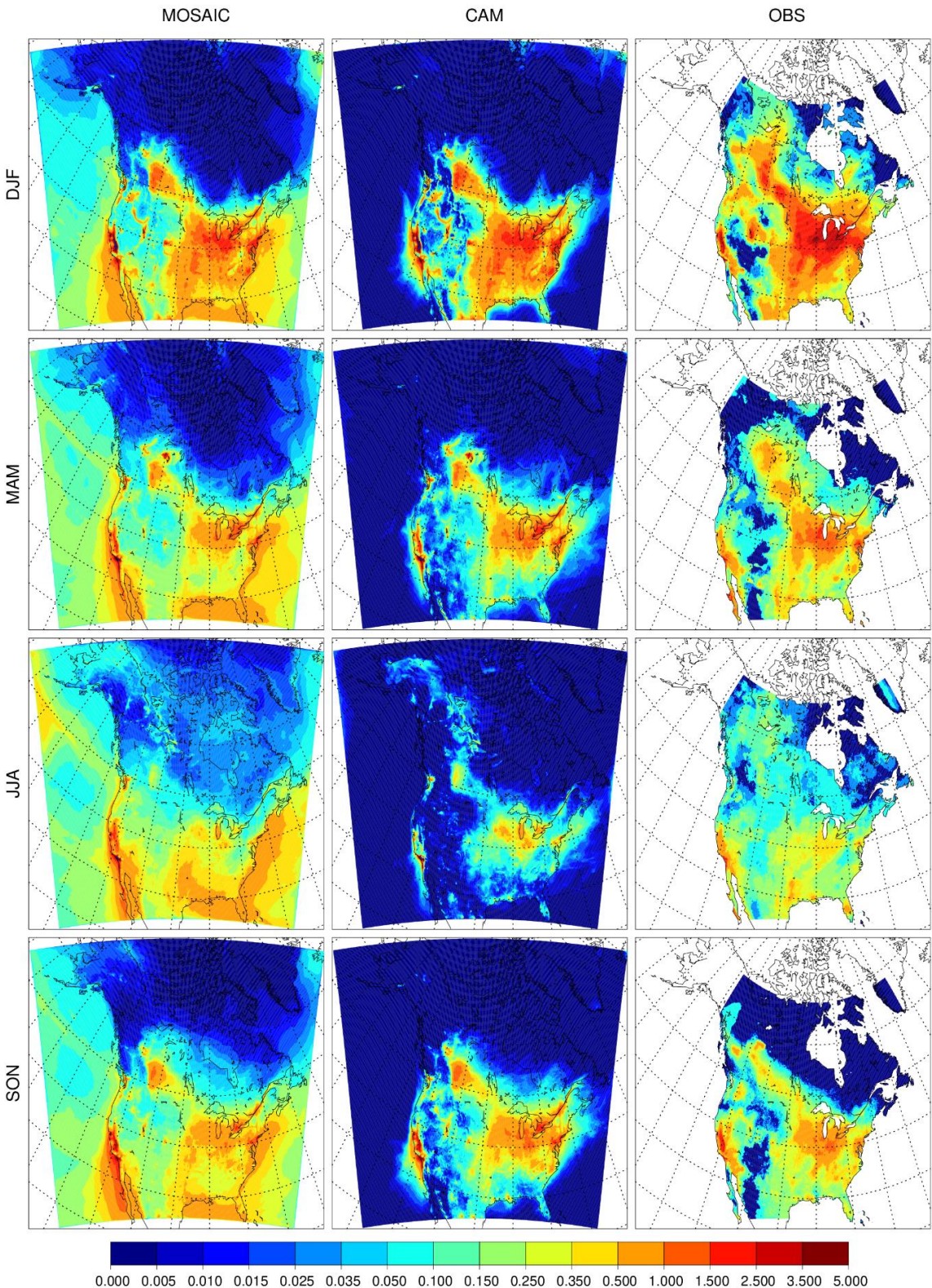

**Figure 1b: Same as Fig. 1a but for nitrate.**

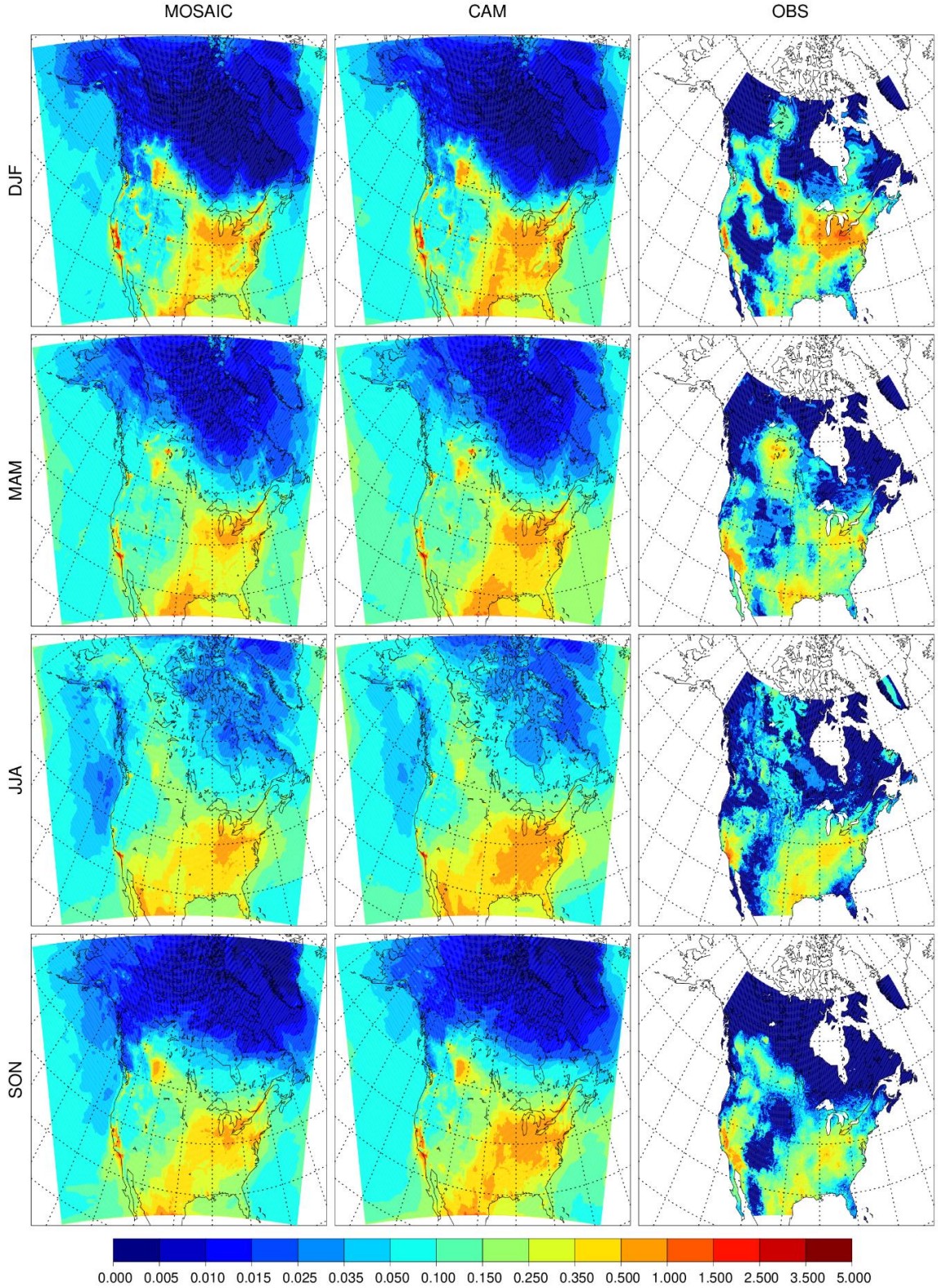

**Figure 1c: Same as Fig. 1a but for ammonium.**

The lower MOSAIC ammonium concentrations in the Great Lakes region and to the south are in agreement with reduced nitrate in this region when compared to CAM.   However, this is not the case for Florida and the Gulf of Mexico coastal region where higher nitrate is associated with lower ammonium.   This is consistent with nitrate being transported inland from production regions over sea water where substantial ammonia emissions are absent in the model.

The impact of changing the dry deposition scheme on sulfate is shown in Figure 2a.   The low bias relative to CAM seen in the REF simulations essentially disappears in the EMR simulations and MOSAIC has slightly higher values compared to CAM in summer and fall.    The relative difference between the EMR and reference simulations is shown in the right two columns.    MOSAIC exhibits an increase over most of North America in all four seasons. By contrast, CAM exhibits a general decrease outside of Alaska in spring, summer and fall.   Both aerosol schemes

show a decrease in all seasons over the US south-east.   The opposite response of MOSAIC and CAM over most of the year to dry deposition changes reflects the size distribution difference between these options in GEM-MACH which impact dry deposition (see Section 4.3).

Figures 2b and 2c show the EMR nitrate and ammonium, respectively.    The excess nitrate compared to the observation product over Florida for MOSAIC is substantially reduced.   There is general attenuation of MOSAIC

nitrate in coastal regions in all seasons except for Baja California and the US north-east.   CAM shows a similar pattern but weaker in magnitude.    MOSAIC also shows a more pronounced change compared to CAM in the continental interior.    The change is negative in JJA and SON but is positive in DJF and a mixed picture in MAM. The EMR nitrate change is positively correlated with the EMR ammonium change in DJF and in the Great Lakes region in MAM but is negatively correlated in other seasons and in the western USA.

Nitrate and ammonium have distinct size distributions in the MOSAIC runs with nitrate occurring more in the larger size bins (see Fig. 11) with ammonium peaking in the accumulation mode.  As discussed further in Section 4.3, the EMR runs involve enhanced removal for particle diameters above 500 nm but reduced removal for smaller particles. In DJF, the MOSAIC nitrate size distribution is dominated by the accumulation mode.   In the other three seasons, nitrate mass is distributed primarily in the coarse mode.   This accounts for the negative sign of the EMR difference

from the reference case in DJF.    CAM lacks the pronounced nitrate peak in the coarse mode, but the overall distribution is shifted to larger sizes compared to MOSAIC (Fig. 11).   This explains the smaller magnitude change relative to the reference case.    By contrast, ammonium mass for both aerosol models is distributed mostly in the accumulation mode throughout the year and the EMR induced difference is positive.

The surface dry deposition flux of PM2.5 sulfate is shown in Figure 3a for the REF runs and Figure 3b for the EMR

runs.  MOSAIC has more dry deposition than CAM in the reference case as seen in the absolute difference (right panels) in spite of having lower sulfate concentrations in winter, summer and fall.    The spring response is distinct but consistent with the monthly progression of the difference with the minimum (defined as spatial extent of the positive difference) occurring in April (not shown).    This behavior cannot be explained as a model spin-up transient.    For the EMR case, the situation is reversed, and CAM has more dry deposition than MOSAIC.   Spring

is not an outlier like in the REF case but has the largest spatially integrated difference of all the seasons.    It is

apparent that with Zhang et al. (2001) dry deposition parameters there is more aerosol removal at the surface for MOSAIC than for CAM.

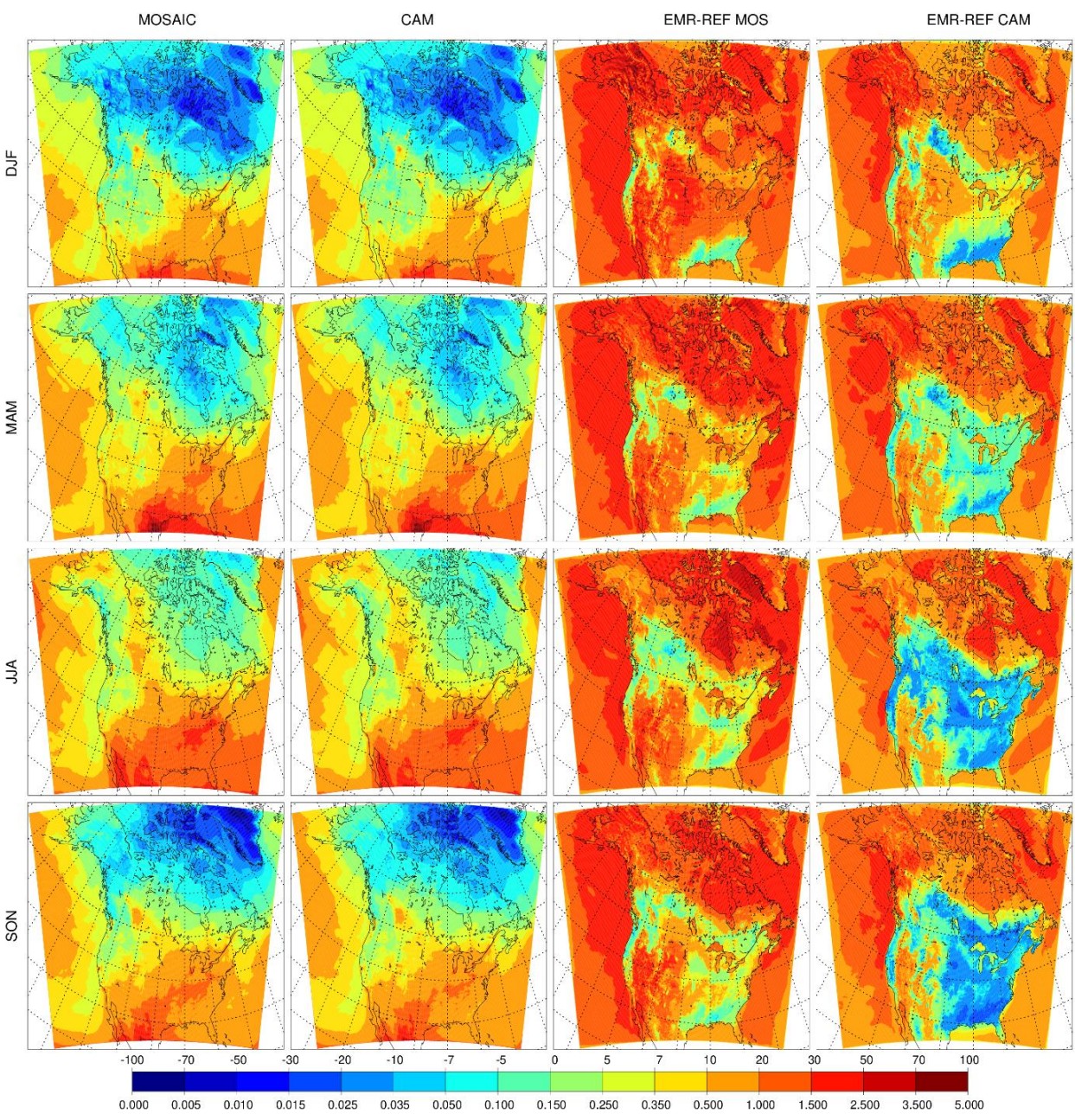

**Figure 2a: Seasonal mean surface sulfate (µg/m³) for the Emerson et al. (2020) dry deposition parameter runs with MOSAIC (first column), CAM (second column) and the difference relative to the reference run from MOSAIC (third column) and CAM (fourth column).**

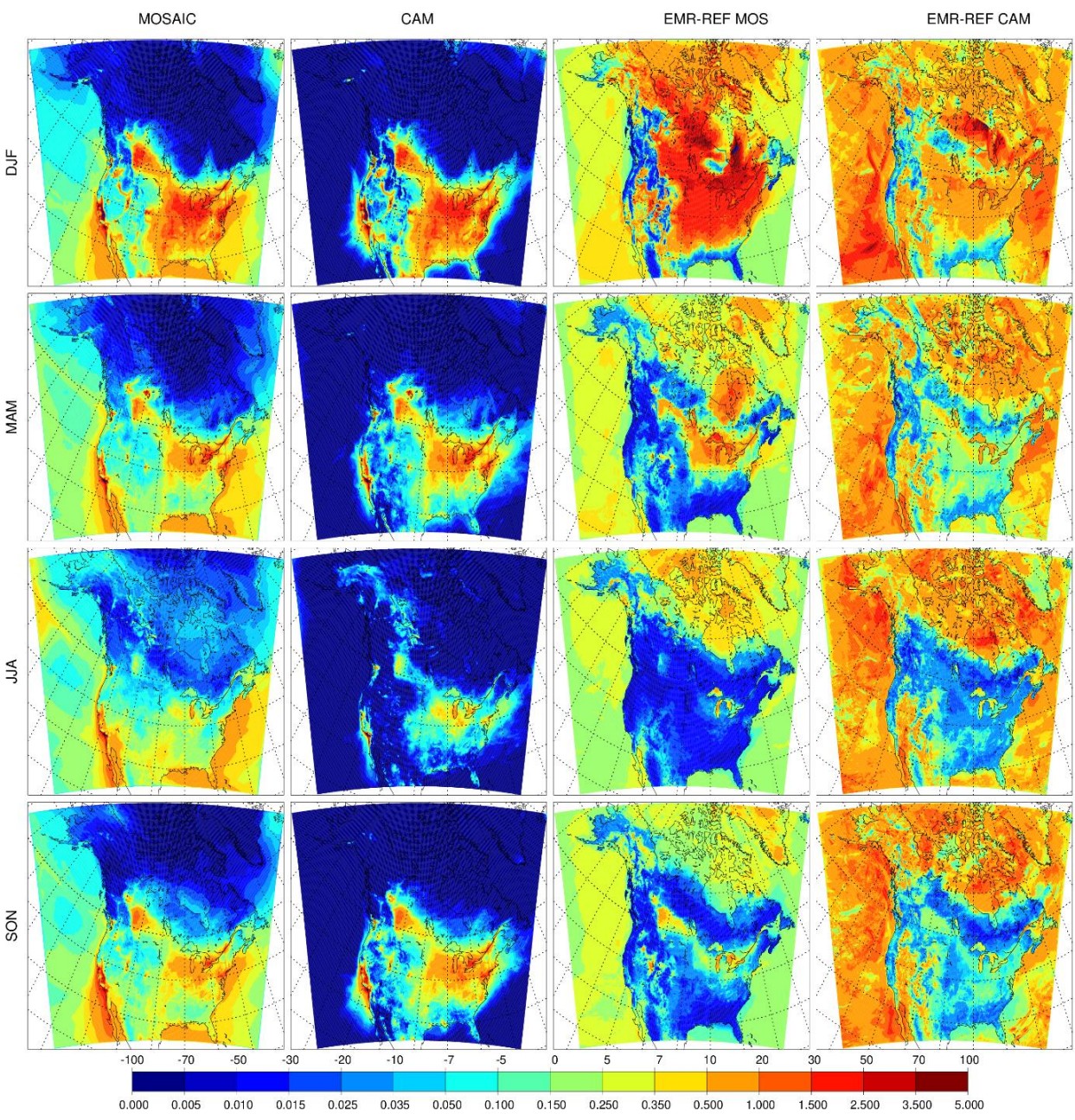

**Figure 2b: Same as Fig. 2a but for nitrate.**

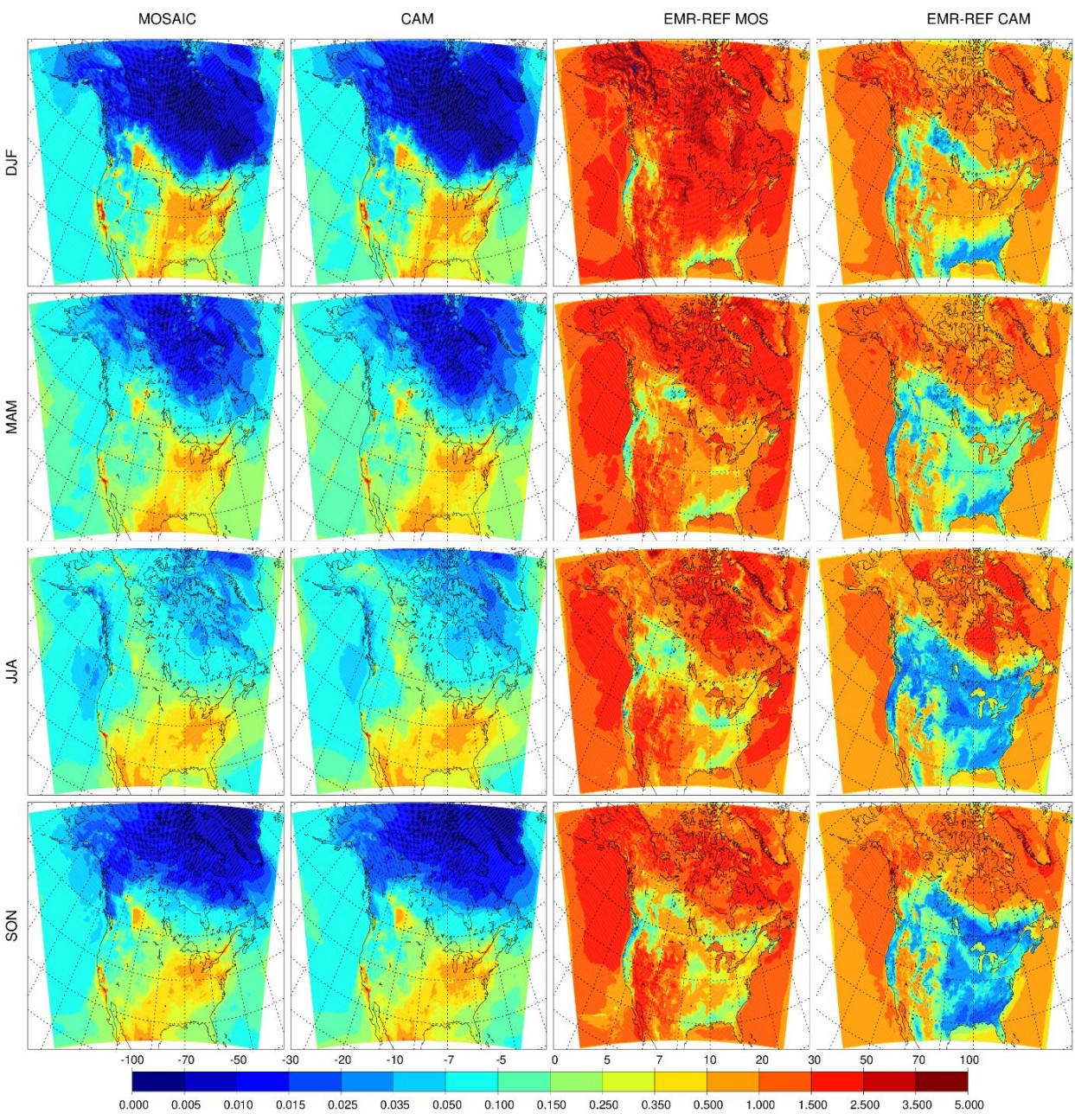

**Figure 2c: Same as Fig. 2a but for ammonium.**

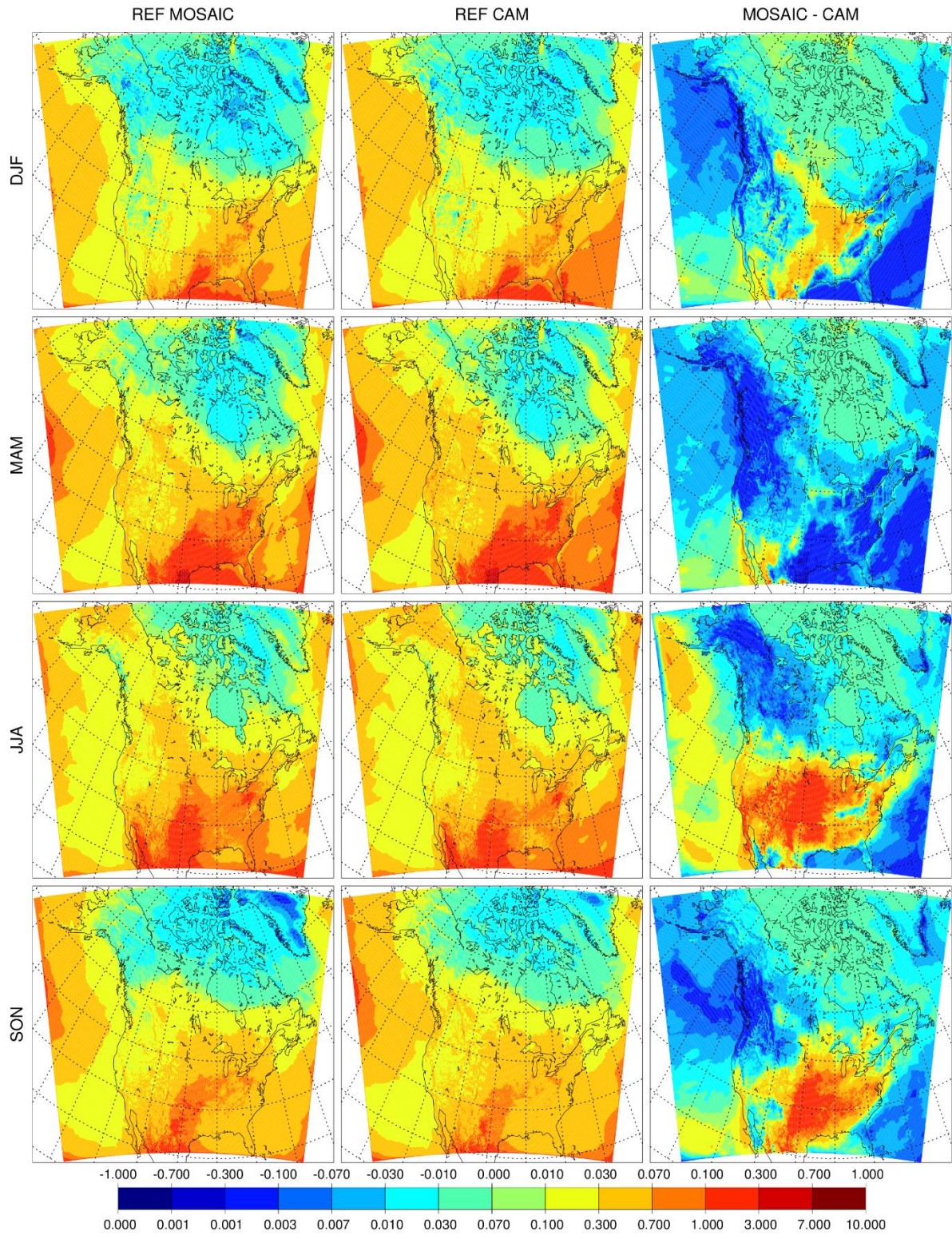

Figure 3a: REF sulfate surface dry deposition flux (μmol/m²) for MOSAIC (left), CAM (center) and the difference (right).

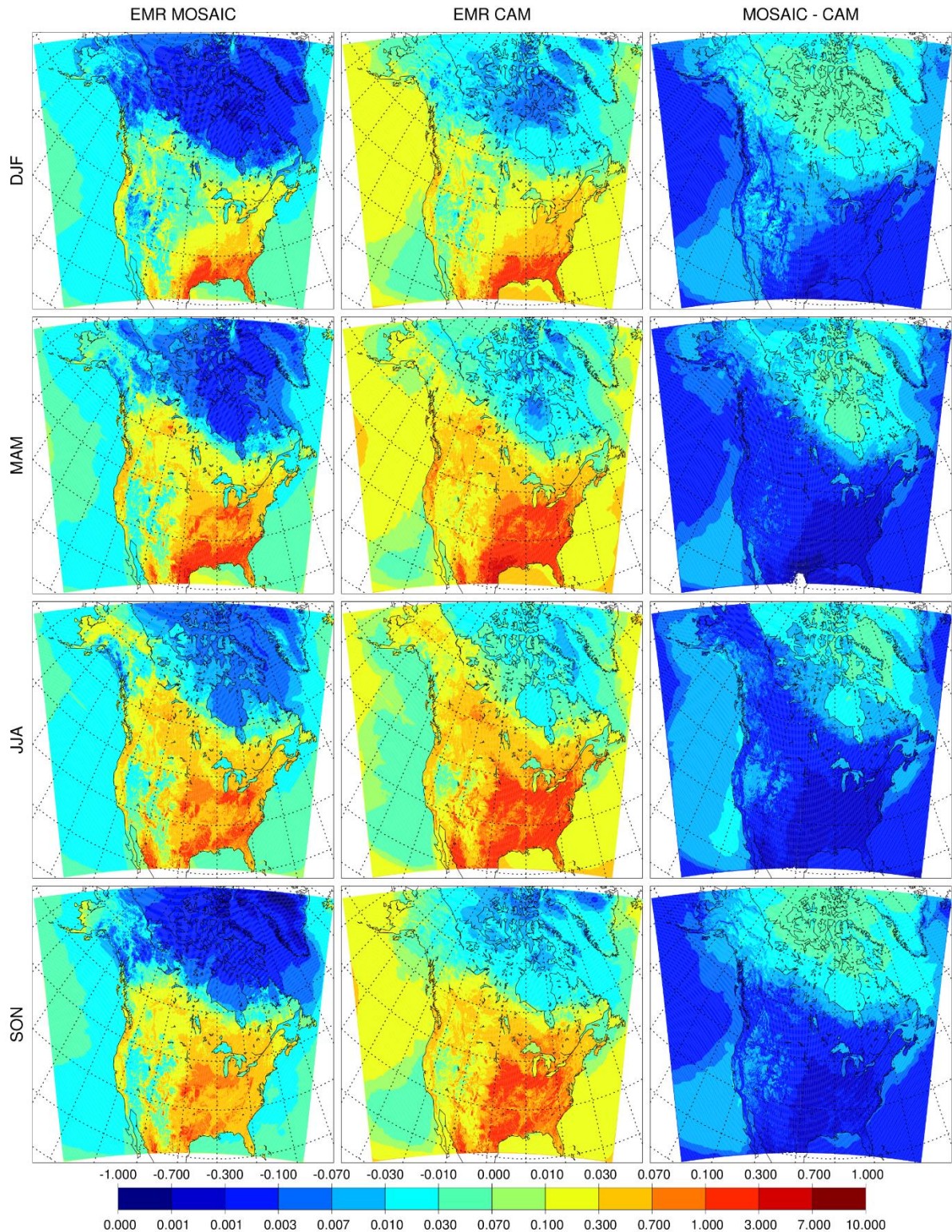

**Figure 3b: EMR sulfate surface dry deposition flux (μmol/m².**

**4.2 Observation Station Network Comparison.**

A monthly evaluation of the model was conducted using aerosol and gas measurement station networks. The model
PM2.5 speciated aerosol mass is compared to three networks: the US Environmental Protection Agency Chemical Speciation Network (CSN), the US Interagency Monitoring of Protected Visual Environments Network (IMPROVE) and the Canadian National Air Pollution Surveillance Network (NAPS). CSN and IMPROVE data can be accessed at https://aqs.epa.gov/aqsweb/airdata/download_files.html, and NAPS data at https://data-donnees.az.ec.gc.ca/data/air/monitor/national-air-pollution-surveillance-naps-program. We group stations from all
three networks into four regions: NA1 for western North America (359 stations), NA2 for the south-eastern US (274 stations), NA3 for the north-eastern US and eastern Canada (346 stations), and NA4 for the Great Lakes and mid-western regions (267 stations) (Figure 4). These regions are approximately the same as defined in Im et al. (2015) with the addition of more NAPS stations and the extra region NA4.

Total PM speciated aerosol model output is compared with data from the Canadian Air and Precipitation Monitoring
Network (CAPMoN) (https://data-donnees.az.ec.gc.ca/data/air/monitor/monitoring-of-combined-atmospheric-gases-and-particles/major-ions-and-acidifying-gases/) and the Clean Air Status and Trends Network (CASTNET) (https://gaftp.epa.gov/CASTNET/CASTNET_Outgoing/data/). These networks cover rural areas. As with PM2.5 observations we group these stations into the North America domain and the four subregions defined above. The focus here is on aerosols but a more detailed comparison of $NH_3$, $HNO_3$, $SO_2$, $O_3$ and $NO_x$ is given in the
Supplement. Measurement uncertainties for the observation networks considered here are given in Table S3.1 in Section S3 of the Supplement. In general, the CSN, IMPROVE and NAPS concentrations of sulfate and nitrate have uncertainties around 15% or less. For CAPMoN and CASTNET the sulfate uncertainty bound is 9-13% and for nitrate it is 18-25%. For ammonium the uncertainty bound is under 15% for networks except for CSN where it is 20%.

Figure 5 shows the North America domain annual time-series of monthly all-station average of sulfate, ammonium and nitrate for 2016 for CAM (blue), MOSAIC (red) and observations (black). Rural (middle row), urban (bottom row) and combined (top row) station concentrations are shown. Solid curves are for REF CAM and MOSAIC runs. The dashed curves are for EMR runs. Curve labels include values from two similarity metrics. The first of the pair is based on curve length (Cao and Lin, 2008) and the second is based on area between curves (Jekel et al., 2019).
Comparison is relative to observations and lower values correspond to a closer fit. The first metric captures shape and can have similar values even though the spread between two curves is different so the second metric acts to separate such cases. The similarity metrics from Figure 5 are tabulated in Table S4.1 with some additional discussion in Section S4 of the Supplement.

The REF MOSAIC sulfate shows a low bias relative to CAM in the 10-20% range as expected from the surface
distributions presented in the previous section. MOSAIC sulfate is also low relative to observations for most months. The combined station sulfate for both aerosol options shows a pronounced low bias relative to observations in winter months and April. This is true for both the REF and EMR runs. This model bias is driven from rural station locations, where there is also an underprediction in May, June and November.

The REF CAM combined station nitrate is closer to observations in the summer months compared to MOSAIC, but

this reflects an offsetting of a high bias in urban stations compared to a low bias in rural stations. MOSAIC has a high summer-time bias for both rural and urban stations. As with sulfate, there is a low bias in winter for both CAM and MOSAIC dominated by the rural station locations. Ammonium from MOSAIC is reduced compared to CAM and is closer to observations for both urban and rural stations. However, both CAM and MOSIAC have a high $NH_4$

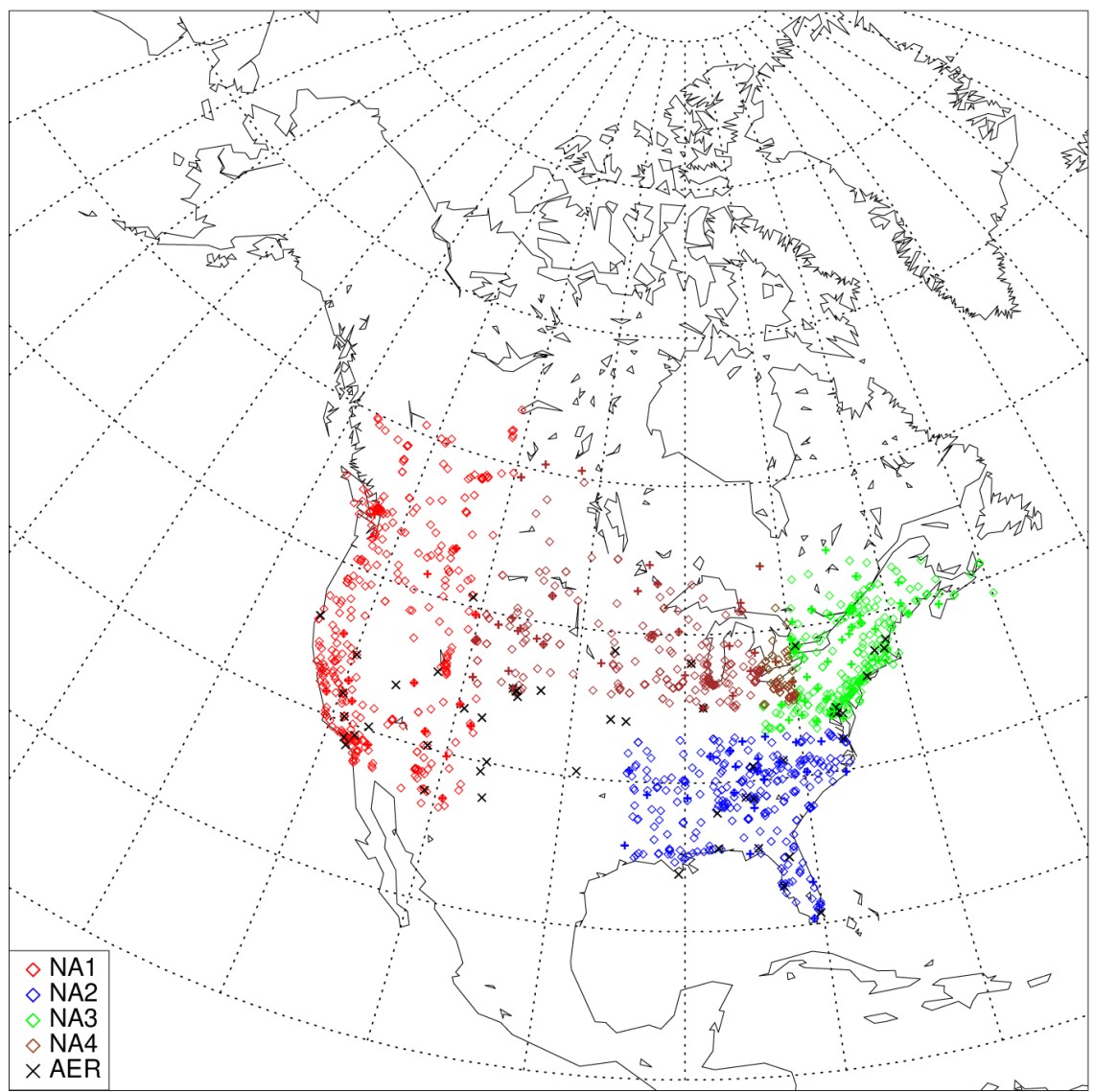

**Figure 4: Locations of speciated PM2.5 and total PM observation stations grouped by region and AERONET stations used for this study (black symbols). CSN, NAPS and IMPROVE stations are represented by diamonds and CASTNET and CAPMoN stations are represented by crosses.**

bias throughout the year. The overall lower ammonium values in MOSAIC is associated with the presence of base cations (Na and Ca) which affect the partitioning of ammonia into the aerosol phase (Makar et al., 2009; Guo et al.,

2018). It is likely that the non-equilibrium characteristics of the MOSAIC scheme are contributing to some extent

as well. The semi-volatile ammonia cannot be assumed to be in steady state at all locations since emissions, transport and loss processes are all time-varying in the model.

The impact of the EMR dry deposition on MOSAIC combined station sulfate (Fig. 5) is substantial with the low bias compared to the reference CAM essentially disappearing. In late fall and winter, the MOSAIC EMR run produces higher combined station sulfate than CAM due to contribution from urban stations. The impact on the CAM

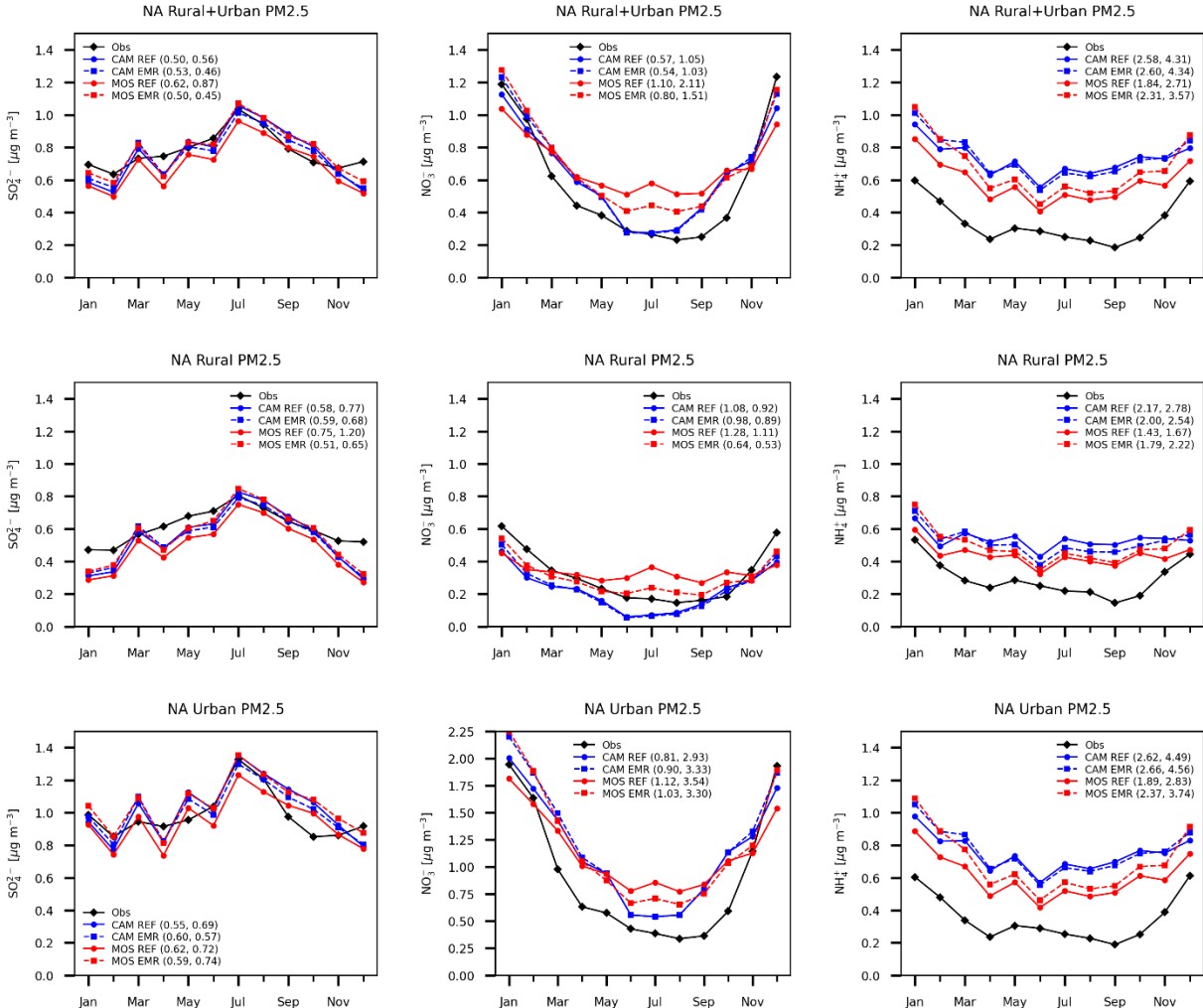

**Figure 5: Station and monthly mean PM2.5 sulfate (left), nitrate (center) and ammonium (right) for combined urban and rural (top row), rural (center row) and urban (bottom row) locations. Sulfate and nitrate are for CSN, IMPROVE and NAPS stations. Ammonium is for CSN and NAPS stations. CAM results are shown with blue curves (reference run solid and Emerson run dashed) and MOSAIC results are shown in red. Observations are shown in black. The first number after curve labels is from a curve length similarity metric relative to observations. The second number is from an area similarity metric. Smaller values mean a closer fit.**

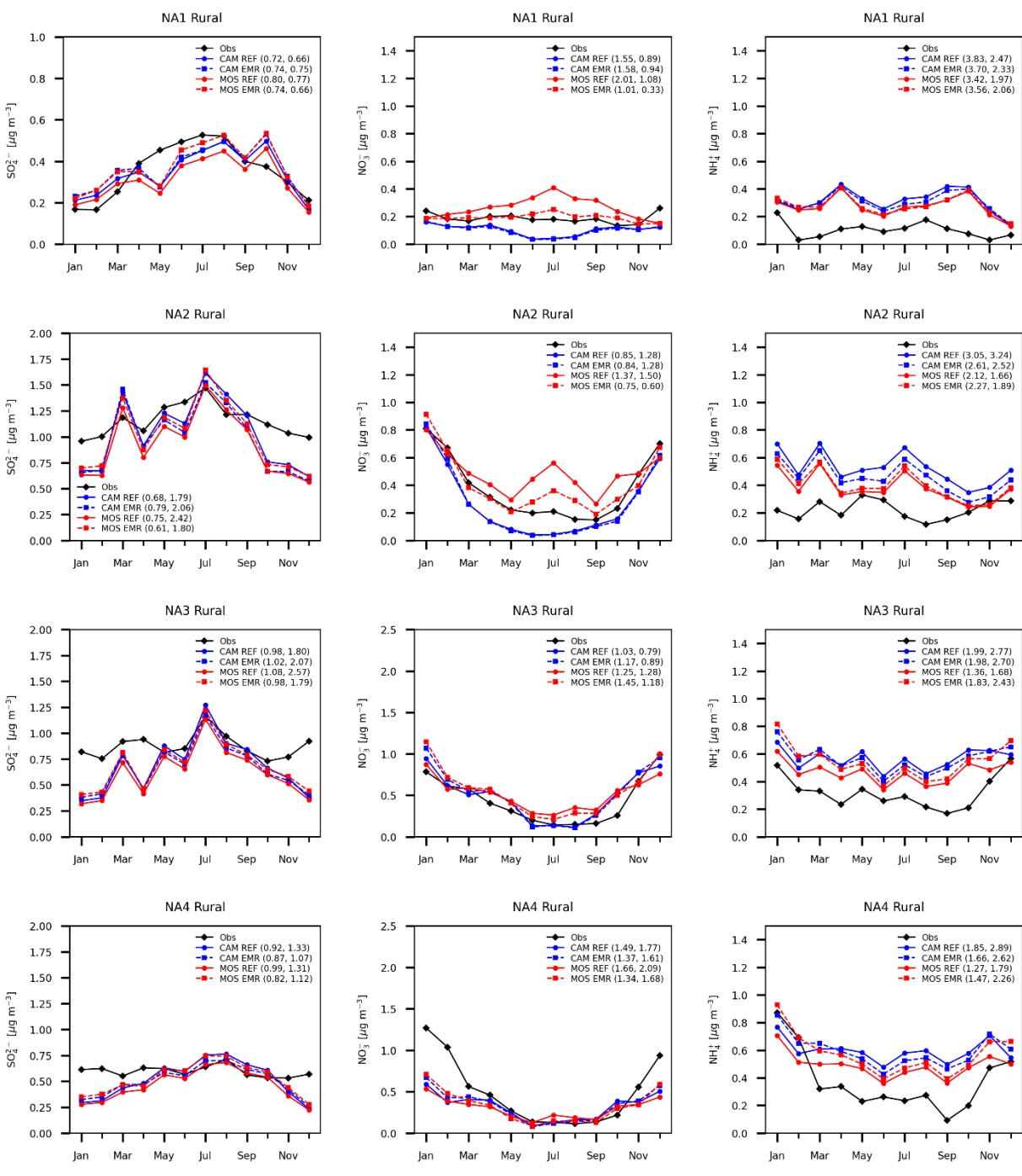

**Figure 6: Monthly mean rural station average sulfate (left), nitrate (center) and ammonium (right) for four North America subregions. Station networks used are the same as for Figure 5. Units and curve options as in Fig 4.**

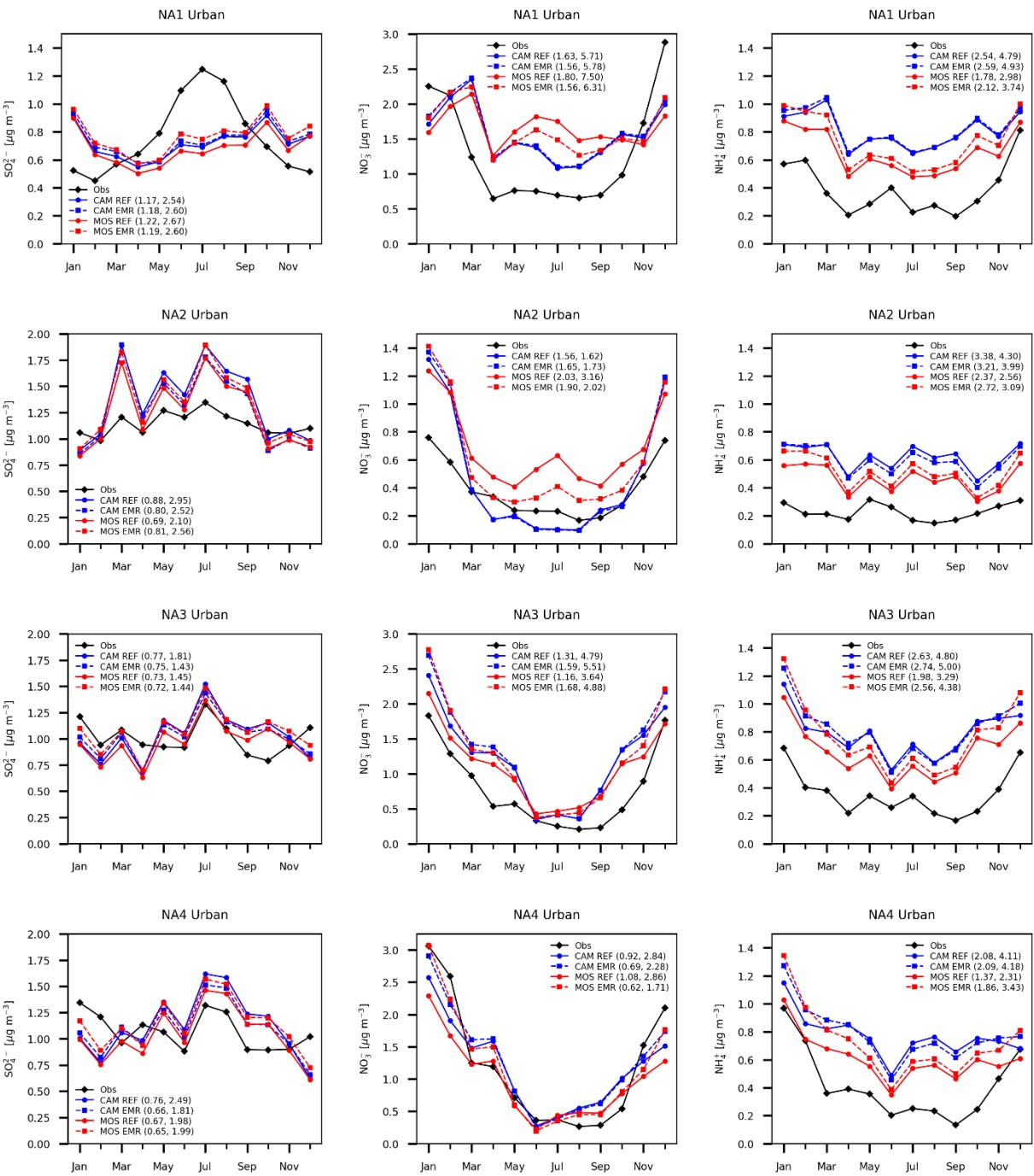

**Figure 7: Same as Figure 6 but for urban stations.**

combined station sulfate is smaller with a shift to higher values before May and to lower values in the rest of the
year. There is a general convergence towards MOSAIC values compared to the REF results for both urban and
rural stations. This is reflected in the reduced shape similarity metric value for EMR MOSAIC and increased value
for EMR CAM compared to REF simulations. Rural stations show the least spread between all four simulations.

The MOSAIC rural station nitrate high bias in summer reduces substantially with the EMR parameters. There is also a reduction in the low bias in winter.   For urban stations the MOSAIC nitrate high bias is reduced noticeably in summer but increased in January and February.   In December, there is good agreement with observations.   CAM sees almost no impact from May to October, with a similar response pattern to MOSAIC in the winter months.   In the case of ammonium, the EMR run shows an increase of the MOSAIC high bias relative to observations compared to the REF case throughout the year with the largest increase in the winter months.   The CAM ammonium undergoes a small decrease, except in winter months, when there is a small increase.   Rural stations show the least spread between the simulations. The ammonium follows the sulfate changes for both CAM and MOSAIC.   The opposite sign of the ammonium change for CAM as opposed to MOSAIC reflects the sign of the sulfate change in the EMR simulations.

A regional breakdown of the station evaluation is shown in Figure 6 for rural stations and Figure 7 for urban stations.   We first consider the REF simulations for sulfate, nitrate and ammonium.   The EMR simulations are considered subsequently.   In region NA1, the REF sulfate (top row) for both aerosol options follows the seasonal progression in rural locations but fails to do so in urban locations.   In urban locations there is too much sulfate in winter and parts of spring and fall, but too little in summer.   Region NA1 is distinct from the other three regions (NA2-4) in that model sulfate is biased high in winter instead of being biased low at urban locations.   Region NA1 also stands out as having the smallest winter bias out of the four regions at rural locations.   The model has a distinct late fall, winter and early spring low bias in sulfate at rural locations in the other regions.   The model follows the sulfate seasonal cycle at rural locations in regions NA2-4 between May and September.

The sulfate from the western boundary over the Pacific Ocean (Fig 1a) has some impact on region NA1 due to the prevailing westerly winds which are weaker in summer compared to winter.   Region NA2 is affected by inflow from the southern boundary condition over Mexico and the Gulf of Mexico, but not in winter (Wang et al., 1998). The low sulfate bias in winter in region NA2 reflects the continental air mass during this period.   The low sulfate bias in spring and summer at urban locations in region NA1 appears not to be related to Pacific Ocean inflow of low sulfate air considering the high level of agreement between model and station observations at rural locations.   The model has a high bias for $SO_2$ in region NA1 (see Fig. S13 in the Supplement) which points to a deficit in conversion to sulfate in the model.   By contrast, region NA2 has excessive sulfate during this period which may be linked to inflow from the southern boundary condition (Fig 1a).   Regions NA3 and NA4, which include numerous polluted urban locations, do not exhibit the same degree of bias.

The REF CAM nitrate in region NA1 (Fig. 7, middle row) is closer to observations at urban locations in the May to September period compared to MOSAIC.   But the model has a high bias from March to October regardless of the aerosol scheme. At rural locations MOSAIC has a high bias as opposed to a low bias in CAM.  Both MOSAIC and CAM exhibit a low bias in winter at rural locations which is more pronounced at urban locations.   Region NA2 has a similar pattern of spring to fall MOSAIC high bias and CAM low bias at rural locations    However, at urban locations CAM has a low bias during this period unlike in region NA1. In winter both MOSAIC and CAM have a large high bias at urban locations.   The winter nitrate bias pattern is opposite that of sulfate.  In region NA3 both

aerosol schemes follow the observations well with a small spread between different simulations at rural locations but with a general high bias and more spread throughout the year at urban locations. The low spread between different simulations at rural locations is also true for region NA4 but with a large low bias in winter and parts of spring and fall which is distinctive from the other regions. At urban locations a noticeable low bias is found during this period as well.

In the case of ammonium (Fig. 6 and 7, bottom row), MOSAIC and CAM have a high bias in the REF runs in all four regions at both rural and urban locations. Region NA1 has the smallest difference between aerosol options at rural locations. MOSAIC is generally closer to observations than CAM in all four regions, especially for urban locations. The winter high bias in ammonium at urban locations appears to correlate with the high bias in nitrate except in region NA1. For rural locations regions NA2 and NA4 fail to show a correlation between high ammonium and nitrate values in winter in the model. In general, the observations do show a correlation between ammonium and nitrate over the course of the year. Region NA2 is an exception at both rural and urban locations.

The reduced bias relative to observations for MOSAIC is associated with reduced uptake of ammonia on account of the presence of base cations from crustal material emissions and sodium from sea salt which is not accounted for in HETV in CAM. Nevertheless, there is a high ammonium bias independent of aerosol scheme which could reflect excessive emissions of ammonia. However, AQ models such as GEM-MACH tend to have a low bias in $NH_3$ prompting adoption of schemes to deal with bi-directional fluxes (Whaley et al., 2018; Pleim et al., 2019). The simulations presented here did not include bi-directional fluxes and surface $NH_3$ is biased low compared to observations in all four regions throughout the year (see Fig. S13 in the Supplement). Another explanation for the bias is missing effects of organic species. Liggio et al. (2011) find that uptake of organic vapors by sulfate aerosol can inhibit the uptake of ammonia. Organic coatings on aerosol particles inhibit the uptake of ammonia as well (Silvern et al., 2017). GEM-MACH aerosols are assumed to be internally mixed with no core-shell structure and organic constituent thermodynamics are not included, so these effects are not captured.

Next, we consider the EMR runs. The MOSAIC run shows an increase in sulfate relative to the REF simulation in every month in all four regions for both rural and urban stations. For CAM, the picture is more complex. The overall change is less than for MOSAIC and includes decreases in addition to increases at different times of the year. There is a tendency for the EMR MOSAIC and CAM curves to converge towards each other. The EMR sulfate change does not substantially impact on the seasonal progression relative to observations. In region NA1 there is an increase in the winter high bias (particularly in urban locations) even as there is a reduction of the winter low bias in regions NA2-4.

For nitrate, CAM has almost no change between the REF and EMR runs in regions NA1 and NA2 compared to substantial changes for MOSAIC. In regions NA3 and NA4 there is a more visible increase in winter and spring for CAM but the change in MOSAIC is much smaller. There is a tendency for the CAM and MOSAIC results to converge towards each other, especially in winter, in all four regions and at both rural and urban locations. Regions NA1 and NA2 are influenced by ocean air inflow and MOSAIC has substantial nitrate formation over ocean water due to sea salt sodium acting to form $NaNO_3$. The EMR dry deposition increases removal of the upper end of the

PM2.5 size distribution which is affected by sea salt emissions (see section 4.3). CAM has no cation mediated
nitrate formation over ocean water and thus has the lowest concentrations there. These low nitrate concentrations
result in depressed concentrations inland in regions NA1 and NA2.

In the case of ammonium, the change between the REF and EMR runs is of opposite sign for CAM and MOSAIC in
the April to October period at both rural and urban locations. CAM has a drop in ammonium whereas MOSAIC
has an increase. In winter both CAM and MOSAIC exhibit an increase relative to the REF case except at rural
locations in regions NA1 and NA2. For MOSAIC there is an increase in the high bias relative to observations
throughout the year with the largest increase in the late fall, winter and early spring period. The spread between
CAM and MOSAIC for both the REF and EMR cases is largest at urban locations. This is associated with urban
cation emissions (e.g. road dust) which reduce uptake of ammonia with MOSAIC (Guo et al., 2018). But these
cations are unable to remove the high ammonium bias in the model.

Additional diagnostics in the form of PM2.5 correlation scatter plots (daily average values in monthly bins) are
given in the Supplement. The monthly scatter distributions of sulfate are very similar for CAM and MOSAIC
(Figures S1, S4, S7 and S9) which is consistent with the same formation scheme being used for both models. Some
reduction in the low bias in MOSAIC is apparent for the EMR case. For nitrate, the CAM REF and EMR runs show
a skew towards low values (and the observation axis) in the March to November period as concentrations decrease
towards 0.1 $\mu$g/m$^3$. This feature is mostly absent in MOSAIC (Figures S2, S5, S8 and S11). By contrast, both
CAM and MOSAIC show excessively high values of ammonium in every month at low concentrations compared to
observations (Figures S3, S6, S9 and S12). This suggests that potential interference from organics in $NH_4$ uptake is
most prominent in relatively unpolluted environments. The model low bias in nitrate can at least be partly explained
by missing $N_2O_5$ hydrolysis. The improvement with the MOSAIC option appears to be linked to the fact that it
produces substantially more nitrate in bin 8 compared to CAM (Fig. 11), which contributes to PM2.5. The anti-
correlation with $NH_4$ in the low concentration limit is misleading. Nitrate amounts are linked to excess ammonium,
after formation of ammonium sulfate, in the whole concentration range as expected based on analysis at individual
station locations (not shown).

The distinct sensitivity of CAM and MOSAIC to the dry deposition scheme is due to differences in the size
distribution of aerosol constituents which we consider in the next section. The size distribution is also involved in
the distinct response of PM2.5 nitrate to ammonium changes in regions NA1 and NA2 compared to regions NA3
and NA4. In the latter, the nitrate change between the REF and EMR runs in winter is much larger than in the
former. This pattern is evident in Fig. 1b (two right columns).

The total PM mass sulfate, nitrate and ammonium are shown averaged over month and the CASTNET and
CAPMoN station locations in Figure 8. These stations represent background or rural concentrations. As with rural

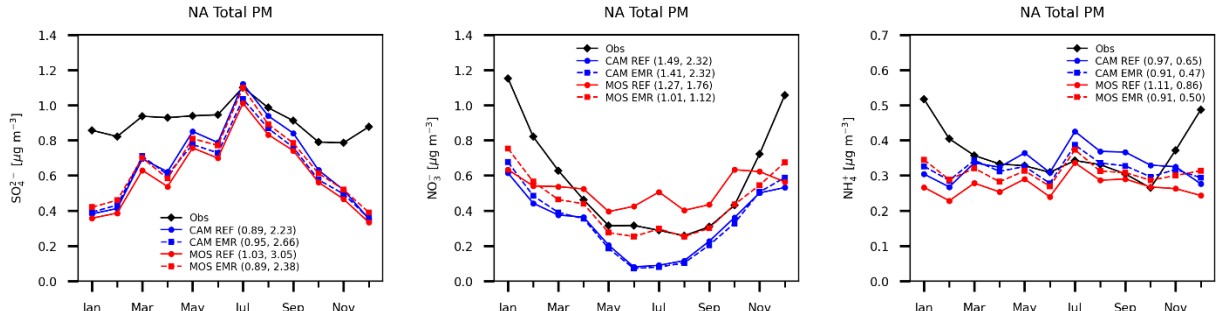

**Figure 8: Model monthly and station mean total PM mass for sulfate (left), nitrate (center) and ammonium (right) at combined CASTNET and CAPMoN locations. CAM results are shown in red, MOSAIC results are in blue and observations are in black.**

PM2.5 sulfate (Fig. 5), the EMR run has MOSAIC and CAM results converge towards each other. For MOSAIC there is a small improvement relative to observations throughout the year but for CAM there is a degradation from May to November. By contrast to the rural PM2.5 sulfate (Fig. 5), the low bias is substantially greater for total sulfate for every month aside from July indicating a large deficit of coarse mode sulfate independent of the model aerosol option. Compared to rural PM2.5 nitrate the total nitrate shows a large low bias in fall, winter and spring.

The MOSAIC EMR run lies closer to observations compared to CAM with very good agreement between April and October. For MOSAIC the REF simulation high bias from spring to fall is removed by the EMR dry deposition scheme but there is little change for CAM. By contrast, for ammonium CAM EMR results show a reduced high summer bias and low winter bias relative to observations. The MOSAIC EMR run shows a reduced low bias for months excluding October. Unlike the rural PM2.5 case there is no degradation of the ammonium results relative to

observations for the MOSAIC EMR run. The total ammonium does not exhibit the high bias seen in rural PM2.5. If the CSN, IMPROVE and NAPS stations sample similar air masses, then this can indicate a deficit in the coarse mode, primarily in winter. However, it is possible that the model fine mode contribution at CAPMoN and CASTNET station locations does not exhibit the same high bias (Fig. 6).

Figure 9 shows the speciated total PM mass for the four subregions defined previously. The MOSAIC EMR results

show a reduced sulfate low bias relative to observations for most months in all four subregions in contrast to CAM EMR results which show an increased low bias. The MOSAIC EMR run shows a substantial reduction relative to the REF run high bias in nitrate from spring to fall in regions NA1, NA2 and NA3 which are subject to marine air mass inflow. The lack of the base cation nitrate formation pathway in CAM results in negligible difference between the EMR and REF runs. For ammonium the EMR dry deposition scheme results in reduction of the

MOSAIC low bias for most of the year in all four regions. In the case of CAM, the EMR run shows a reduction of the high bias from spring to fall. This difference pattern indicates that the fine mode contribution to the total mass does not have the same high bias seen at CSN and NAPS rural stations (Fig. 6) where the ammonium high bias increases for MOSAIC due to reduced dry deposition of the fine mode with EMR parameters.

All aerosol model options, independent of dry deposition scheme, show a total PM low bias against observations in

winter. Region NA1 has the smallest difference in January and February. The other three regions show a much

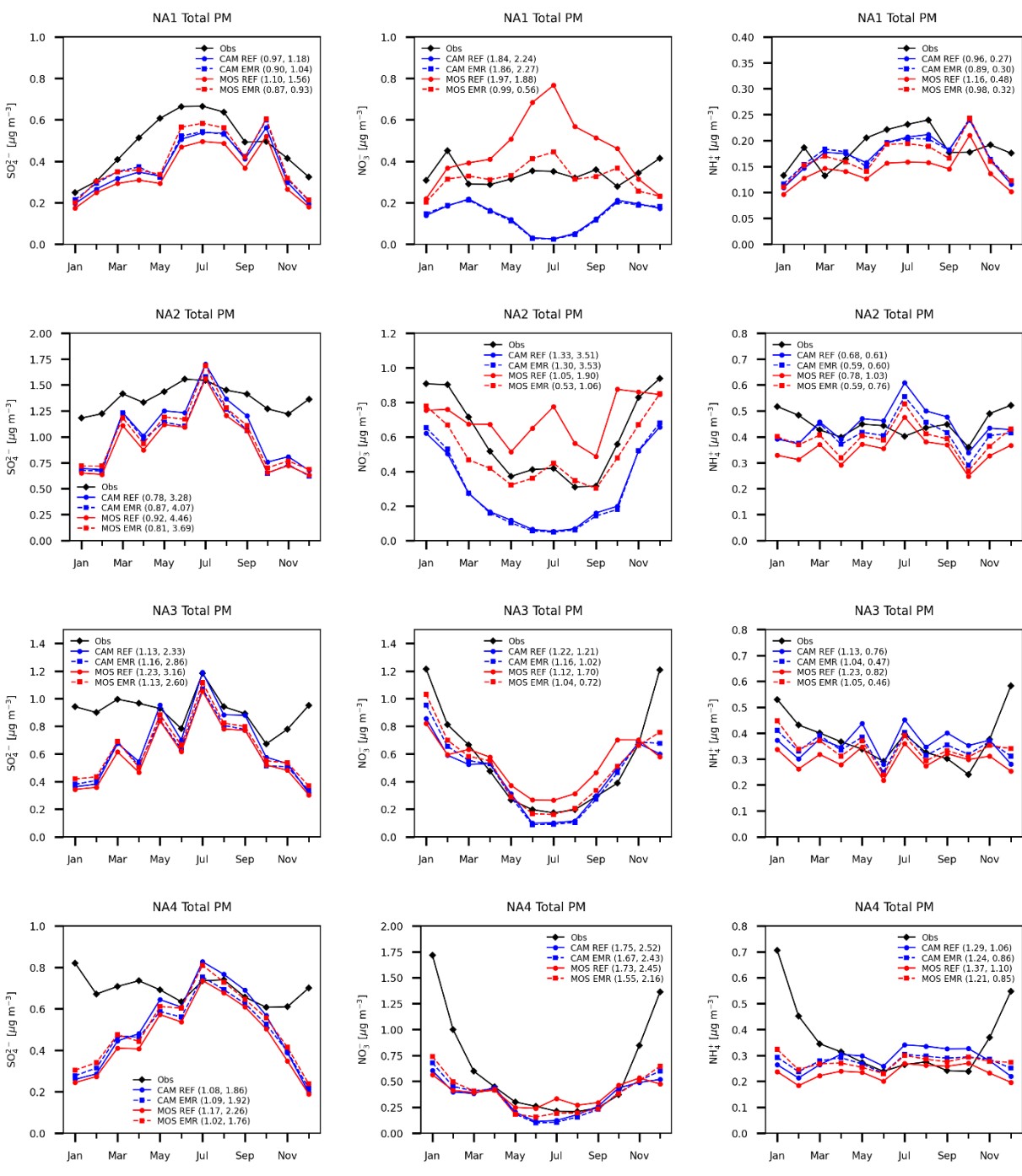

**Figure 9: Model monthly and station mean total PM mass in four subregions as in Figure 6 but for the combined CASTNET and CAPMoN stations.**

larger difference compared to region NA1. The winter low bias for total PM nitrate and ammonium in all four regions is distinct from the high bias seen in PM2.5 aside from region NA1.

The impact on $NH_3$, $HNO_3$, $SO_2$, $O_3$ and $NO_x$ (defined as $NO + NO_2$) of the aerosol options is presented in Supplement Section S2. The REF simulation particle nitrate picture is consistent with the response of $HNO_3$. Increased MOSAIC surface level nitrate is associated with decreased nitric acid gas concentrations (Figure S13). In regions NA1 and NA2, the MOSAIC nitric acid can be up to 30% lower compared to CAM for some months. By contrast, the difference in $NH_3$ between MOSAIC and CAM is very small in all four regions. Nitrate concentrations are comparable to $HNO_3$, but ammonia concentrations are typically much higher than particulate ammonium. There is almost no impact of the aerosol option on $SO_2$, $O_3$ and $NO_x$. The model lacks aerosol phase heterogenous reaction pathways that can affect the concentration of these species (e.g. Yang et al., 2024; Lou et al., 2014). Sulfate formation is affected by aqueous chemistry in cloud water and the availability of $H_2SO_4$ which is produced in the gas phase, and neither pathway is affected by the aerosol phase in the model (clouds and aerosols are not coupled). The model lacks pathways to convert $HNO_3$ back to $NO_x$ (e.g. Zhou et al., 2003) and aerosol phase processes do not impact $HNO_3$ formation in the model (such as heterogeneous conversion of $N_2O_5$ into $HNO_3$).

The effect of changing the dry deposition scheme on the gas phase constituents is very small and not shown. Most of the difference is in $HNO_3$ for the MOSAIC option with the EMR case showing slightly higher values than the REF case in all four regions. CAM shows almost no difference between the EMR and REF simulations. As discussed in the next section, the size distribution difference between MOSAIC and CAM simulations impacts $HNO_3$ since MOSAIC shows distinct particulate removal compared to CAM.

## 4.3 AERONET Volume Size Distribution

To evaluate the aerosol size distribution produced by GEM-MACH against observations, we use the Aerosol Robotic Network (AERONET, https://aeronet.gsfc.nasa.gov/) column integrated volume size distribution (VSD) inversion product (version 3). The AERONET VSD has a range of 22 aerosol radius values from 0.05 µm to 15 µm. The model VSD is obtained by vertical integration of the 3D total aerosol volume field derived from the sum of individual constituent mass (including water) divided by the constituent density.

Uncertainty estimates for the AERONET VSD mode parameters (volume median radius and standard deviation) are available (Sinyuk et al., 2020) and have been used to produce size-dependent distribution errors (see Section S5 in the Supplement for details) which are shown in Figure S14. AERONET fine mode size distribution retrievals have been compared with in situ aircraft measurements by Schafer et al. (2019) in several US regions. Generally, the AERONET deviation in the peak concentration radius is 5.2% and in the VSD fine mode width is 15.8%. Roger et al. (2022) created a model based on size distribution parameters from 851 AERONET globally distributed stations for use with satellite measurement validation. The fine and coarse mode particle volume concentrations have an uncertainty under 10% except for 440 nm optical depths less than 0.05. The uncertainty in the particle volume

median radius in the fine and coarse modes is about 0.02 µm and 0.32 µm, respectively. The standard deviation of the size distribution uncertainty is about 0.04 and 0.044 for the fine and coarse modes, respectively. For the data considered here these uncertainties correspond to about 10.5% for the fine mode peak radius and 9.2% for the mode width. For the coarse mode the values are 11.3% and 7%, respectively. These errors are much higher than the provided in the AERONET uncertainty analysis product (see Fig. S14, top left panel). However, the deviation of the model from observations is much larger than the AERONET VSD uncertainties.

The monthly mean AERONET data for 2016 that is used here has substantial gaps due in part to forest fires. We have selected 52 North American stations (Figure 4) where only three months at most are missing. The CAM and MOSAIC run output was sampled with the same gaps. These gaps have no significant impact on the station averaged distributions of the model output if the missing periods are included. Thus, we infer that AERONET data for 2016 is sufficiently representative in temporal terms. Comparison of the annual mean AERONET data for 2017 with substantially fewer gaps shows no large change compared to 2016 in terms of magnitude and monthly variability. To see if there is a spatial sampling issue with the 52 AERONET stations considered here, we compared CAM and MOSAIC average of all aerosol network station locations and found no qualitative change in the size distribution and its monthly variance (not shown). Since model vertical profiles of relevant fields are available only in once-daily snapshots the comparison with AERONET VSD data presented here should be considered qualitative. The 52-station model average does not account for local diurnal variation seen by AERONET. However, we do not expect such diurnal variation to produce substantial differences in the aerosol size distribution.

Figure 10 (left panel) shows the annual mean, vertically integrated VSD for reference simulations with CAM and MOSAIC together with AERONET inversion data. The EMR results show very little difference compared to the REF case and are not shown. The accumulation mode in CAM is shifted towards higher diameters with a peak at just over 500 nm. The AERONET accumulation mode peak occurs at about 300 nm which is captured by MOSAIC. MOSAIC produces a distinct minimum around 850 nm which agrees with the AERONET observations. This minimum is poorly represented in CAM and the coarse and accumulation modes overlap excessively. Both CAM and MOSAIC produce a coarse mode peak at about 5 µm which is smaller than the AERONET peak at about 6 µm. The model substantially underestimates the wet mass in the coarse mode which may account for the peak size bias in this mode. This could involve wind-blown dust which is not included in our model but a low bias is found in models that do and is related to problems with emissions, turbulent transport and dry deposition (e.g. Adebiyi and Kok, 2020). Coarse mode wind-blown dust tends to maximize in spring and summer (e.g. Hand et al., 2019) which is broadly consistent with the model bias. The sea-salt size distribution is expected not to have such a bias   The bias in the coarse mode peak may indicate that coarse mode particulate in the atmosphere has a surface area to volume ratio larger than the homogeneous density spherical particles assumed in the model (e.g. Adebiyi et al., 2023). This will reduce the gravitational settling rate and shift the peak to larger sizes if the mass distribution in the primary emissions is sufficient at those sizes.

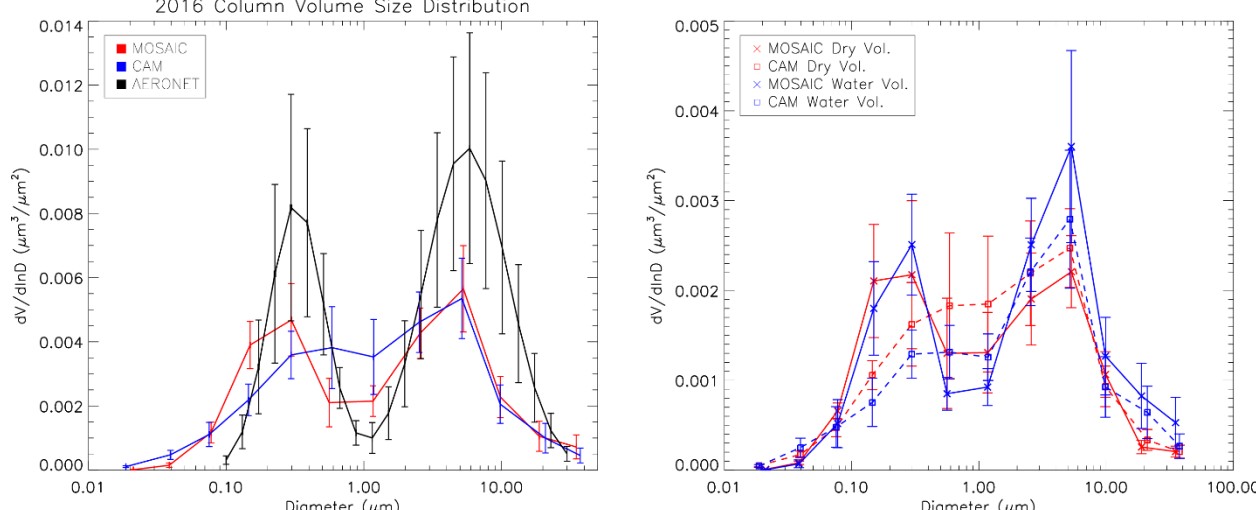

**Figure 10: Annual mean AERONET station average column integrated wet volume (left panel) for reference CAM (blue), reference MOSAIC (red) and observations (black). The right panel shows the fraction of dry mass volume and water volume corresponding to the curves in the top panel. Error bars are monthly standard deviation.**

CAM and MOSAIC underestimate the total wet column volume by about a factor of two in the annual mean. The

675 AERONET VSD shows much larger seasonal variability than the model (see Figure S15 in Section S5 of the Supplement) which is reflected in the monthly standard deviation error bars in Figure 10. The largest deviation of the model from observations is in summer and applies to both the fine and coarse modes. The PM2.5 fraction of wind-blown dust maximizes in spring (e.g. Park et al., 2010) and accounts for under 5% of the mass so it cannot explain the disagreement. The model also has less inter-monthly variation of the VSD in each season (Fig. S15),

which may reflect local aerosol loading not captured in the model. The emissions in the model distribute minor point sources over the grid box as area emissions. The grid resolution for our simulations cannot resolve constituent tracer filaments narrower than 10 km and any such grid-scale filaments would be subject to numerical dispersion. This implies that point sampling in the model always sees a smaller range of values compared to the real atmosphere.

The difference between the model and observations may also involve vertical transport issues. We did not compare model and observational vertical profiles of aerosol mass but CAM and MOSAIC produced similar vertical aerosol distributions at the network station locations (not shown). In addition, the disagreement between the model and station observations at the surface in terms of total aerosol volume is much smaller than the column VSD difference (not shown). This implies that there is a model deficiency in the vertical transport of tracers.

The AERONET VSD uses log-normal distributions for inversion, and this appears to underestimate values in the small size limit of each distribution resulting in a low bias around 0.1 µm and 1.0 µm. This makes comparison with the model around these sizes problematic but should not undermine it over most of the size range.

The contribution of water vapor to the total column VSD is shown in the right panel of Figure 10. Water accounts for about half the VSD in the accumulation mode for MOSAIC and about two-thirds in the coarse mode. The

695 CAM aerosol water content is substantially lower than in MOSAIC in the accumulation mode and the dry fraction

exceeds 60%.  In the coarse mode CAM water is closer to MOSAIC values but the fraction of the total VSD is close to half.  CAM aerosol water exceeds that of MOSAIC in the Aitken mode and accounts for slightly more of the VSD than the dry component.  This may reflect the different sectional adjustment schemes used.  Based on our testing with a box model (not shown), the Jacobson moving-center scheme has the propensity to "ventilate" the smallest particle size bins because of rapid particle growth.

**4.4 AERONET Aerosol Optical Properties**

In this section we compare the model column aerosol optical depth (AOD), single scattering albedo (SSA) and Angstrom exponent (AE) with AERONET observations for the stations considered in the previous section.  A simple Mie scattering scheme based on the code in Appendix A of Bohren and Huffman (1983) is used for model profiles.  This code does not consider composition and mixing state effects and applies to homogenous spherical particles characterized by a volume distribution and water content.  The AERONET AOD and SSA uncertainties are discussed in Section S6 of the Supplement.

Figure S16 shows the monthly mean AOD and SSA for 440 nm, 675 nm, 870 nm and 1020 nm wavelengths for EMR MOSAIC, EMR CAM and AERONET.  Due to the difference in the accumulation mode size distribution, MOSAIC exhibits an increased AOD compared to CAM for all months of the year at 440 nm.  For longer wavelengths MOSAIC shows a decrease compared to CAM from May to November.  This is consistent with the largest disagreement in the accumulation mode occurring in summer (Fig. S15) when a realistic size distribution minimum around 1 µm fails to form with the CAM option.  In the case of SSA, MOSAIC exhibits lower values compared to CAM throughout the year for all wavelengths.  The fit to AERONET values is best in summer months but the seasonal cycle seen in the observations is not captured.  SSA depends on the composition, mixing state and size distribution of aerosols (e.g. Tian et al., 2023). The excessively high values of SSA (over 0.9) in the model during winter indicate that there is a composition and mixing state difference that cannot be captured by the simple Mie scattering scheme we use.  Without a more sophisticated aerosol optical properties model, we cannot assess if there are seasonal biases in the aerosol composition.

The column AE averaged over 440-870 nm is shown in Figure S17.  MOSAIC has a much better fit to AERONET AE compared to CAM.  CAM has values that are too low from April to November.  The AE reflects differences in the size distribution and CAM has a poor fine mode distribution compared to MOSIAC and AERONET from spring to fall (Fig. S15).

**4.5 Model Size Distribution Analysis**

The seasonal mean surface size distributions of REF run $SO_4$, $NO_3$ and $NH_4$ averaged over all ground station locations considered in Section 4.2 are shown in Figure 11.  There is a shift of the accumulation mode peak downward by roughly one bin size in all three constituents when MOSAIC is compared to CAM. For sulfate (left

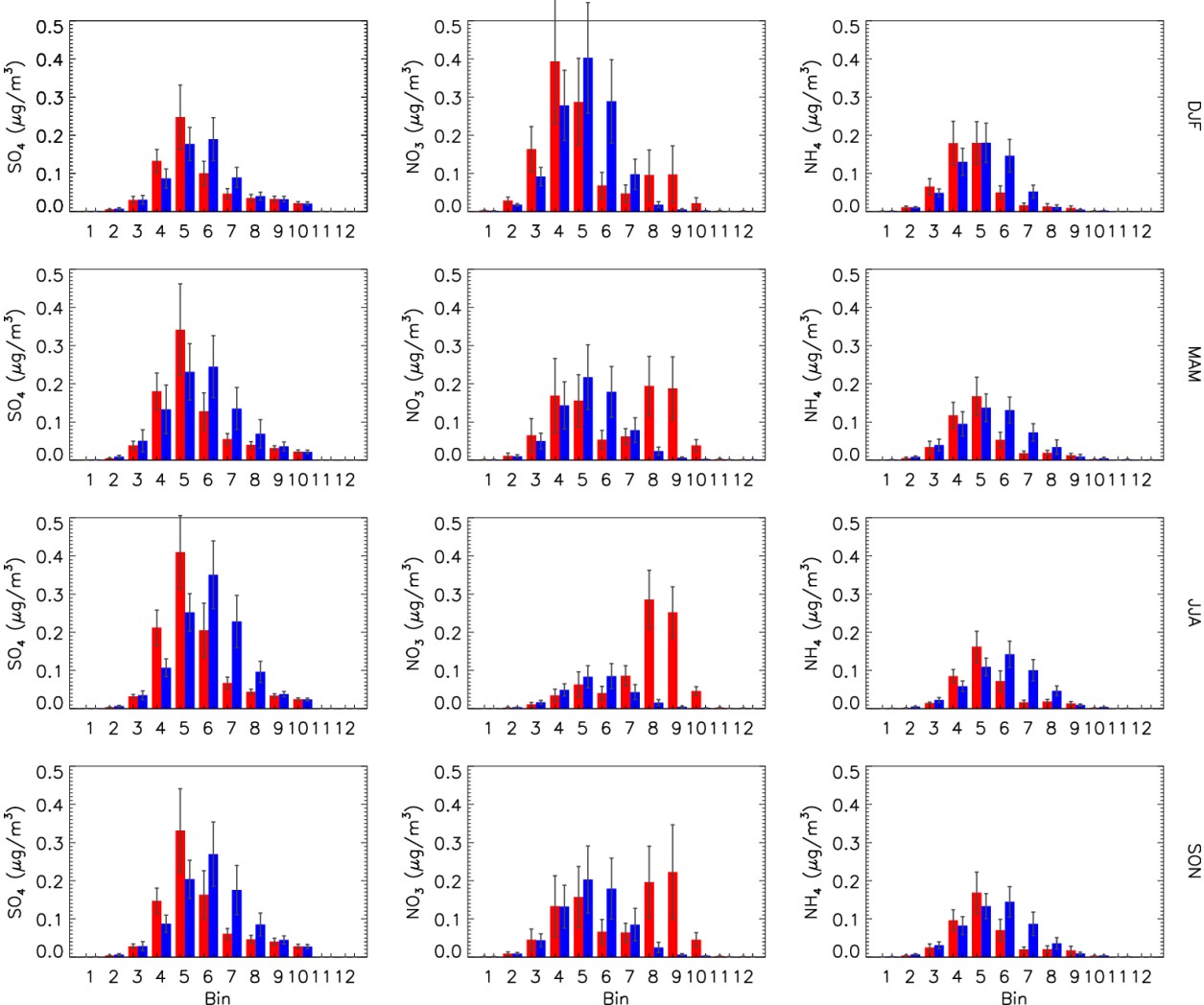

**Figure 11: Seasonal and station location mean size distribution by sectional bin of sulfate (left), nitrate (center) and ammonium (right) for CAM (blue) and MOSAIC (red). Error bars are daily standard deviation. All station locations shown in Figure 4 are used for the average.**

column), the MOSAIC peak occurs in bin 5 compared to bin 6 in CAM. CAM also has substantially higher values in bins 7 and 8 (nominal diameters from 0.64 to 2.56 µm). In the case of nitrate (center column), the MOSAIC accumulation mode peak occurs in bin 4 and in bin 5 for CAM in winter and spring. In summer the CAM nitrate is spread between bins 5 and 6 but MOSAIC has a clear peak in bin 5. In fall both aerosol models produce a peak in bin 5. CAM lacks the distinct nitrate coarse mode produced by MOSAIC in bins 8, 9 and 10. This coarse mode is not reflected in the ammonium distribution (right column). MOSAIC ammonium is shifted to smaller sizes compared to CAM with the peak in bin 5 compared to bin 6 during summer and fall. In winter and spring, the CAM peak occurs in bin 5 but with substantial mass in bin 6. For MOSAIC the peak occurs in bin 5 with the next largest mass amount found in bin 4. In winter the MOSAIC peak straddles bins 4 and 5. This shift appears to be coordinated with the dominance of the accumulation mode peak of nitrate in MOSAIC in winter as compared to the dominance of the coarse mode peak in other seasons.

Averages over coastal stations and continental interior stations (not shown) did not produce any significant difference in the size distribution of nitrate and the other constituents. Thus, the coarse mode nitrate peak is not a feature associated with sea salt sodium. Sodium and calcium are emitted as parts of crustal material. MOSAIC can form $NaNO_3$ and $Ca(NO_3)_2$ from $HNO_3$ and crustal material emissions are predominantly in the coarse mode.

The model seasonal surface station average distributions of aerosol water, dry mass and the water to mass ratio are shown in Figure 12. For all seasons the MOSAIC and CAM water values in bins 8-10 are very similar (left column). In bins 11 and 12, MOSAIC has more dry mass (middle column) and water compared to CAM, but the ratio is lower (right column). There is a shift in the accumulation mode peak for water from bin 6 in CAM to bin 5 in MOSAIC for all seasons. This includes winter when the dry mass peak occurs in bin 5 and bin 4, respectively. The enhanced water uptake in the accumulation mode in MOSAIC is evident in all seasons primarily in bin 5. There is also enhanced uptake relative to CAM in bins 1 and 2 for every season.

A cursory examination of the dry mass ratios of aerosol constituents between CAM and MOSAIC (not shown) does not offer a simple explanation of the differences. MOSAIC has over three times less ammonium than CAM in bin 1 and 2-3 times less sulfate and nitrate, but it has a higher water content. By contrast in bins 11 and 12 CAM has 4-6 times more ammonium and has a higher water content. The amount of ammonium in bins 4 and 5 is nearly the same with both aerosol models. CAM has 40-60% more nitrate and 10-20% less sulfate but MOSAIC has more water. Unlike CAM, MOSAIC takes into account sulfate and nitrate salts of sodium and calcium, but these are very small fractions of the total mass in the fine mode. A more detailed investigation of the differences between the water uptake schemes in CAM and MOSAIC is beyond the scope of this study.

The shift in mass to smaller bin sizes is also apparent in the aerosol number distribution. Figure 13 shows the relative difference between MOSAIC and CAM. Except for the winter season MOSAIC has more particulate in bins 4 and 5. In every season there is 30-50% less particulate in bins 6 and 7 and up to 20% less in bin 8. For diameters over 2.5 µm, MOSAIC has higher particle number. The largest difference occurs in bins 11 and 12. Bins 1 and 2 have 3-20% less particle number depending on season. Unlike CAM, MOSAIC conserves particle number and the differences at the ends of the size distribution appear to reflect sectional adjustment scheme effects. The moving-center scheme transfers both particle mass and number when the growth in a given size bin is sufficient. The single-moment scheme in CAM only transfers mass. For the sensitive moving-center scheme, there is depletion of number in bins 1 and 2 where the particle growth is large. In the case of bin 12 there is no larger bin for the mass and particle number to go, and it can accumulate to some extent subject to scavenging processes. For MOSAIC a larger particle number reduces the sedimentation rate and dry deposition velocity since the average particle diameter in the bin can be smaller than the simple mean of the bin boundaries as used in CAM. Relative differences in bins 11 and 12 are very large due to the small absolute particle number in these bins.

The impact of the EMR dry deposition parameters is shown as absolute differences relative to the reference run in Figure 14.    For both CAM and MOSAIC there is an enhancement of the accumulation mode for bins 1-5 and a

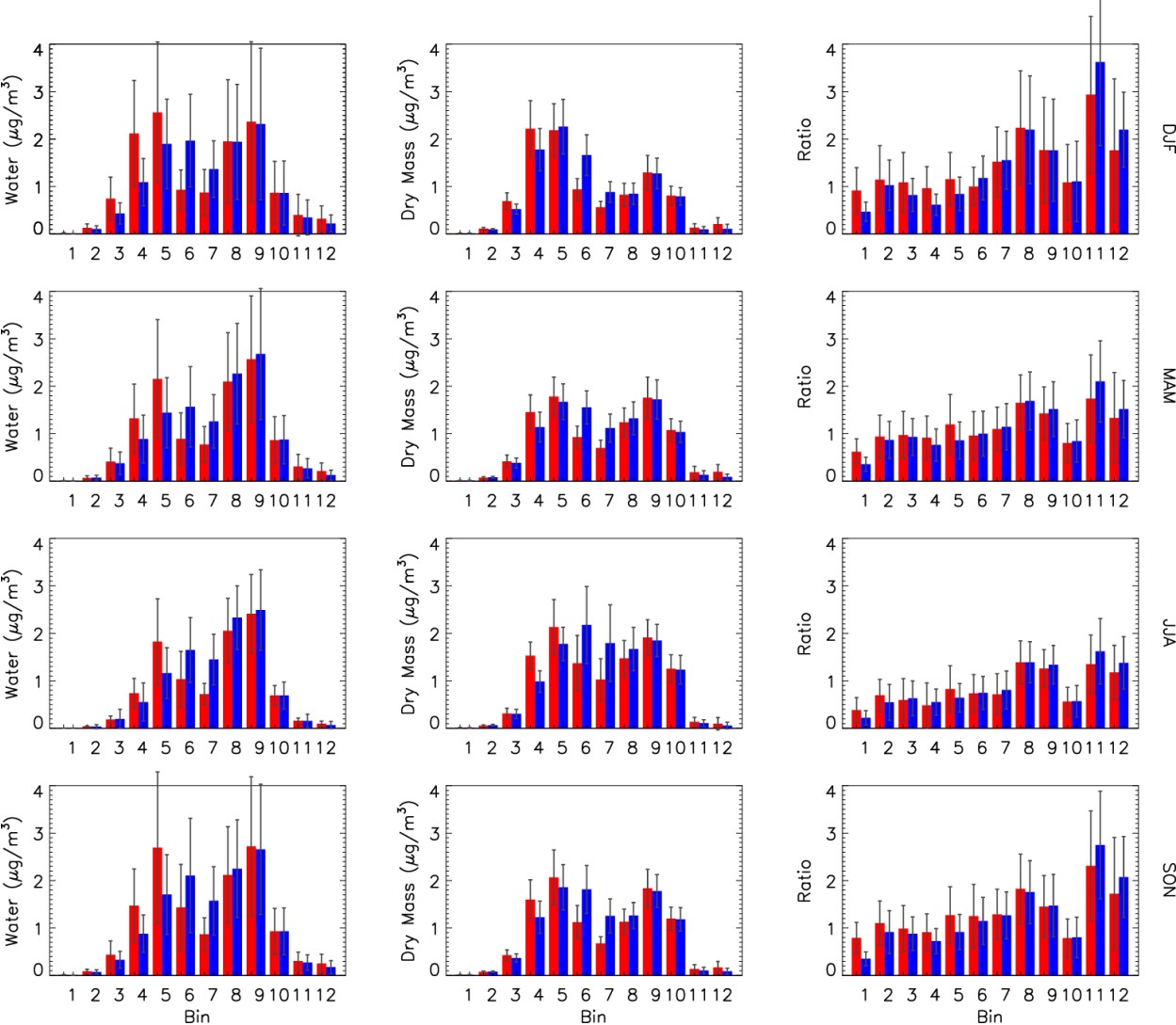

**Figure 12: Seasonal and station mean size distribution by sectional bin of aerosol water (left), dry total mass (center) and the ratio of water to dry mass (right) for CAM (blue) and MOSAIC (red).   Error bars are daily standard deviation.**
**Station locations are the same as in Figure 11.**

decrease for bins 6-10.  The sulfate increase with MOSAIC for bins 3-5 is larger compared to CAM, but CAM has a

larger decrease for bins 6-8.  This pattern is repeated for ammonium but for nitrate MOSAIC has a much larger

decrease in bins 8 and 9, reflecting the large reference values.  The difference in the sensitivity between MOSAIC

and CAM to dry deposition is also apparent in the diagnostics presented in Sections 4.1 and 4.2.    Figure 15 shows

the seasonal all-station average sulfate dry deposition as a function of bin size.  For the EMR case, MOSAIC

experiences the largest drop in deposition in bins 3-5 with smaller changes at coarser bin sizes.  But CAM shows

substantial increases in deposition in bins 6-8.

The surface dry deposition velocity in the Zhang scheme   (see Fig. 1 in Emerson et al., 2020) increases

monotonically with decreasing aerosol size for particle diameters less than 1 µm.  Since the MOSAIC accumulation

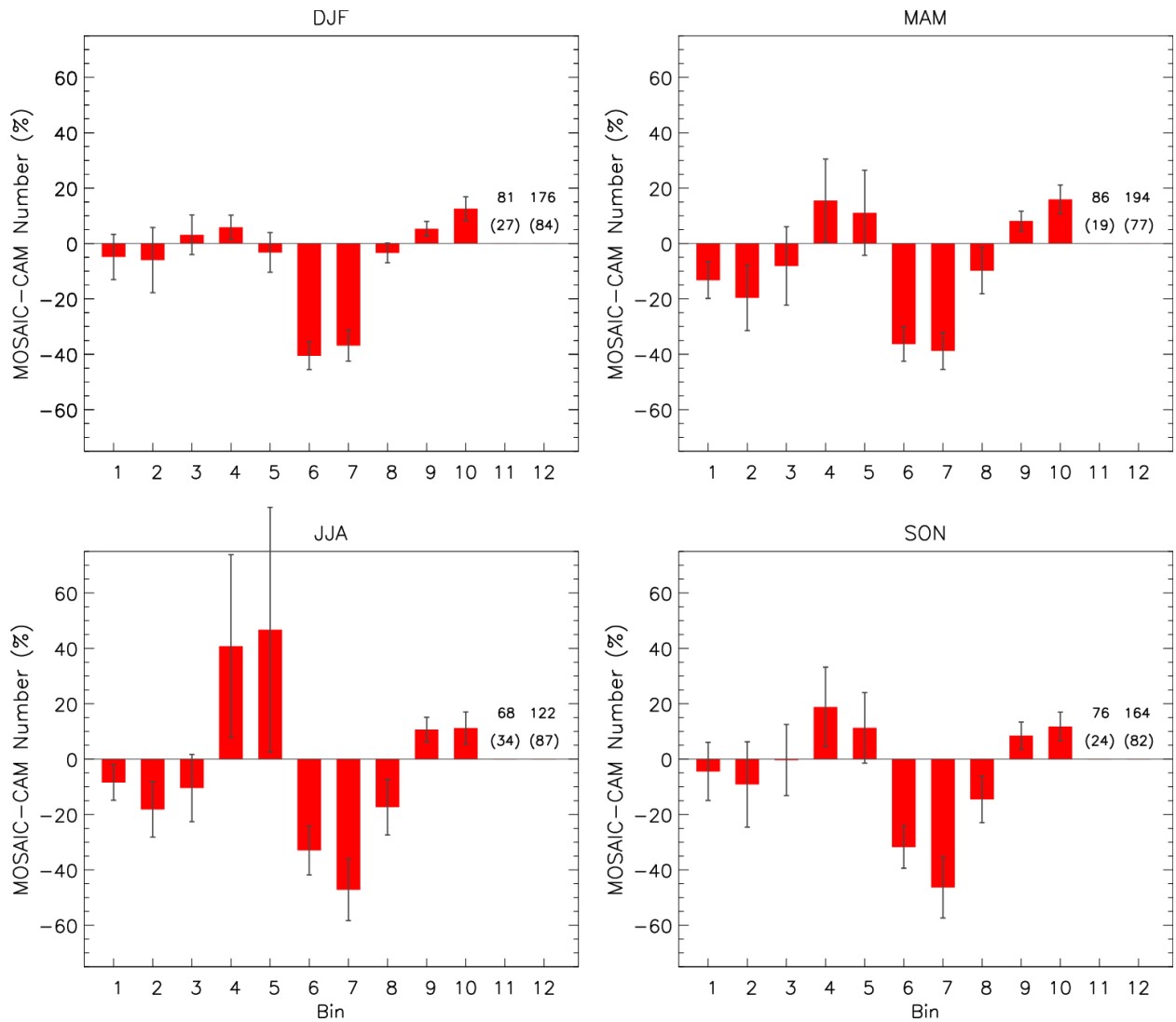

**Figure 13: Relative difference of aerosol number between MOSAIC and CAM for station and seasonal mean distributions. Bins 11 and 12 are shown as value and error in brackets. Error bars are daily standard deviation. Station locations are the same as in Figure 11.**

mode distribution occurs at smaller particle sizes this implies that it is scavenged more than is the case with CAM. With the EMR parameters there is a substantial reduction in deposition velocity in the Aitken and lower accumulation mode (bins 5 and smaller) but an increase in the upper accumulation mode and larger particles up to 20 µm in diameter. The cross-over occurs at around the diameter of 500 nm. This reduces Aitken and accumulation mode scavenging in MOSAIC. However, In the case of CAM there is an increase of scavenging since it has more mass distributed in bins 6, 7 and 8.

The small change in $HNO_3$ between the MOSAIC EMR and REF simulations noted in the previous section reflects the nitrate size distribution. The EMR reduction of dry deposition in the accumulation mode is combined with an increased deposition in the coarse mode due to the nitrate peak in bins 8 and 9 during spring, summer and fall. For

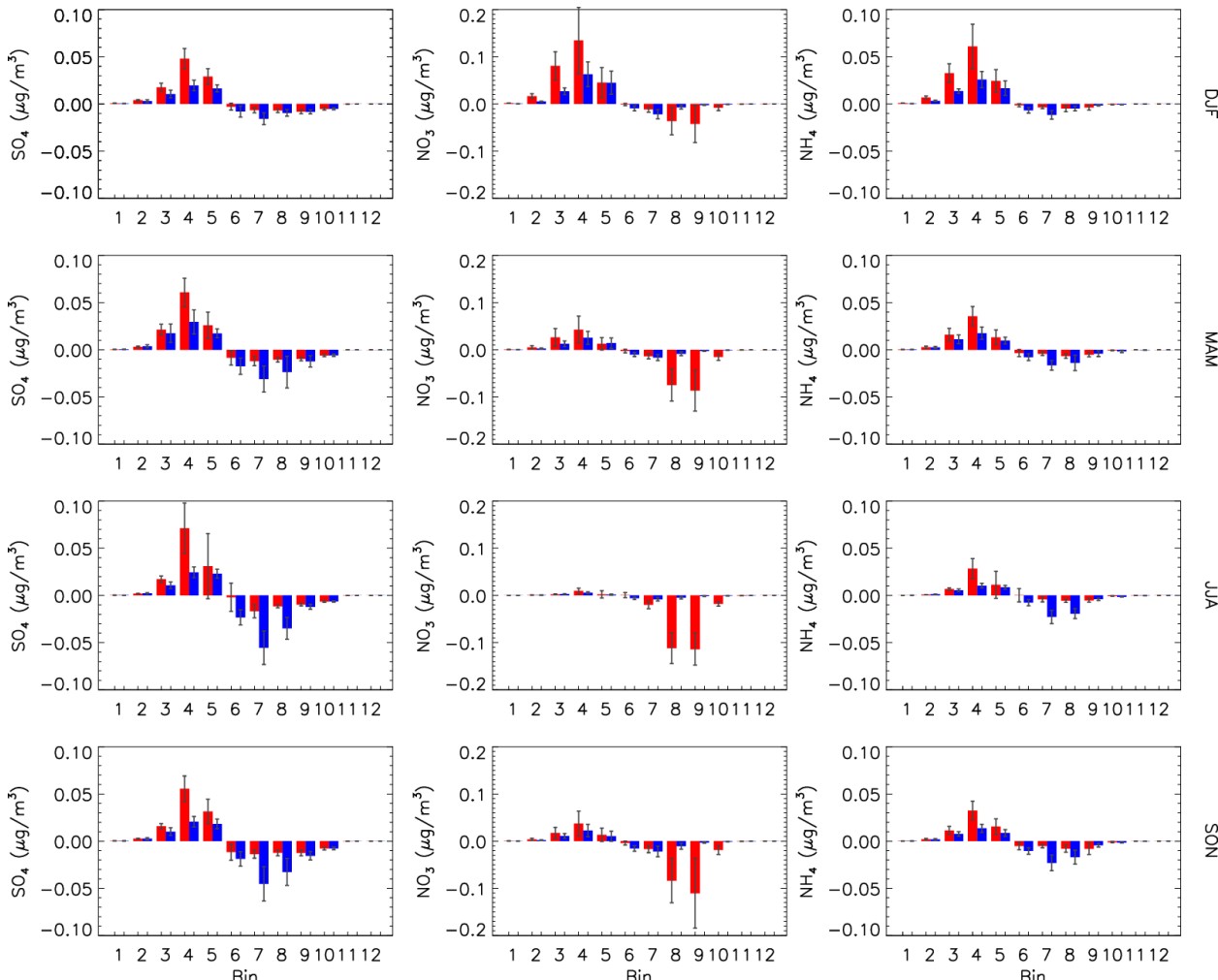

**Figure 14: Seasonal and station mean difference between the Emerson run and reference run (EMR-REF) for each sectional bin for sulfate (left), nitrate (center) and ammonium (right) for CAM (blue) and MOSAIC (red). Error bars are daily standard deviation. Station locations are the same as in Figure 11.**

CAM the nitrate peak spans bins 5 and 6 and the nitrate distribution convolution with the bin-dependent deposition velocity change results in a much smaller impact on $HNO_3$ removal via aerosol uptake as nitrate.

The change in the dry deposition scheme does not affect the poor separation of accumulation and coarse mode peaks for the CAM option. Unlike the case with MOSAIC, where the thermodynamics routine is called for every bin size, in CAM HETV is applied to the bulk composition. The bulk result is distributed to the 12 size bins using Knudsen number dependent weights. To test the impact of this approach, we conducted one-month simulations with HETV called for every size bin (not shown). This resulted in an accumulation mode peak diameter consistent with MOSAIC and AERONET at about 300 nm. However, it did not produce a clear minimum around 850 nm. This points to an issue with the single-moment formulation used in CAM. It is likely that there is too much bin diffusion

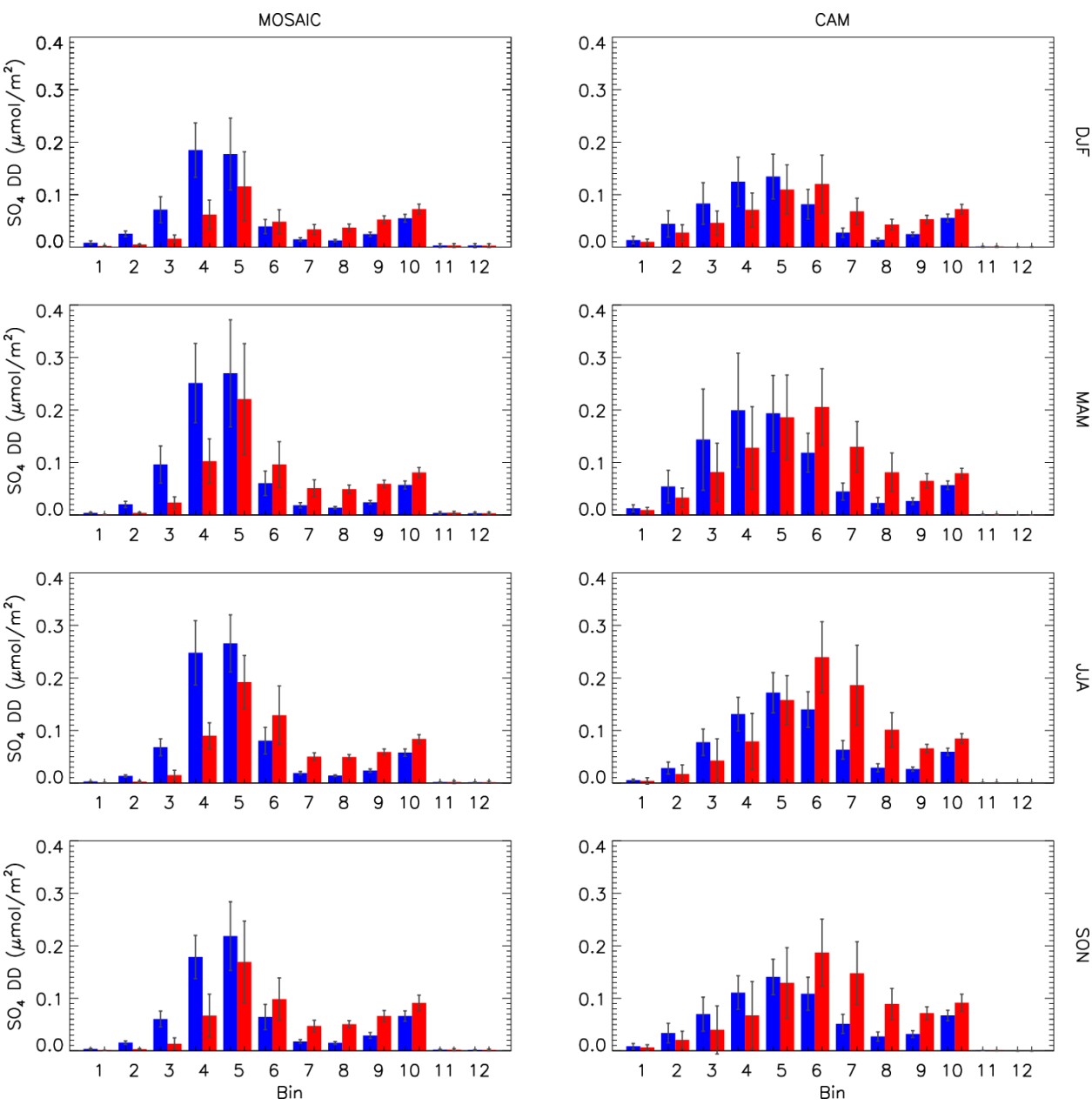

**Figure 15: Seasonal and station mean model sulfate dry deposition size distribution for MOSAIC (left) and CAM (right). Blue corresponds to REF runs and red to EMR runs.   Station locations are as in Figure 11.**

with 12 bins.    The sectional resolution tests in Gong et al. (2003) show that bin diffusion with 12 bins can be substantial and is reduced for higher bin resolution (see their Figure 8a).    The greater bin diffusion for single-moment schemes has been noted by Tzivion et al. (1987) and others.

## 5 Conclusions

We have evaluated the impact on aerosol distributions in GEM-MACH from a comprehensive inorganic aerosol thermodynamics model, MOSAIC.  Compared to observation station network data, MOSAIC offers improvements

in nitrate and ammonium relative to CAM for the reference case. MOSAIC improves the size distribution compared to CAM in the accumulation mode. This is due to the use of a double-moment scheme with thermodynamics applied to each size bin. On account of deficiencies in the Zhang et al. (2001) dry deposition scheme this results in a degradation of sulfate compared to CAM in the reference case. This issue is removed using the Emerson et al. (2020) dry deposition parameters. The MOSAIC impact on aerosol composition derives from the inclusion of base cations which are not present in HETV. We expand on these aspects below.

MOSAIC nitrate shows a tighter correlation with observations (see also the Supplement) with best agreement in rural regions. This can be attributed to the inclusion of base metal cations (Na and Ca). However, the production of nitrate over sea water in the REF simulations is excessive, which affects coastal inland locations. This can be partly attributed to excessive sea salt emissions, but this is not sufficient to explain the bias. Previous work with MOSAIC indicates that the mixing state and associated dependence of uptake coefficients for dust and sea salt aerosol substantially impacts nitrate formation (Wu et al., 2022). Sea salt nitrate is also found to undergo photolytic breakdown much more effectively than $HNO_3$ especially in tropical and subtropical latitudes (Ye et al., 2016; Kasibhatla et al., 2018) and this effect is not included in our model. Another contribution to the high bias is the dry deposition scheme. With the Emerson et al. (2020) dry deposition parameters there is a substantial reduction of the bias in regions which are exposed to marine air mass inflow.

For the reference simulations, MOSAIC has a reduced high bias compared to CAM in PM2.5 and total ammonium in summer months with less improvement in winter months. The MOSAIC ammonium is affected by base cations, and their emissions are reduced in winter for the anthropogenic sources included in our simulations. However, even with base cation effects, the PM2.5 ammonium high bias (most apparent at low concentrations) is not removed which points to other causes. This includes inhibition of $NH_3$ uptake by aerosols due to organic constituent effects not accounted for. This is consistent with the model $NH_3$ being lower than surface observation station measurements with greater partitioning into the particle phase.

Sulfate formation in our study was implemented using the same formulation for both aerosol options but surface concentrations show substantial differences. The performance of MOSAIC relative to observations and CAM is degraded with the use of the Zhang et al. (2001) dry deposition scheme. With the Emerson et al. (2020) scheme MOSAIC performs no worse than the reference case CAM for PM2.5. This is due to the aerosol size distribution, particularly in the sub 1 µm diameter range. CAM differs from MOSAIC in that the thermodynamics routine is not applied to each size bin but instead to the bulk followed by redistribution into bins using weight factors. CAM is also a single-moment scheme that does not conserve aerosol number, which results in higher sectional diffusion compared to the double-moment MOSAIC formulation. These two factors contribute to the poor size distribution in CAM compared to AERONET observations.

Non-equilibrium mass transfer, thermodynamic system solution regimes and treatment of hydration are the primary differences between CAM and MOSAIC. The improved process representation in MOSAIC increases the numerical expense compared to CAM. For the same resolution, domain size and model inputs, MOSAIC takes around three times longer per time-step. The MOSAIC thermodynamics routine is called 12 times at every time-step compared

to once for the bulk HETV treatment. Dynamical mass transfer between the gas and aerosol phase also results in more variance in model time-steps depending on local conditions. A further consideration is that the version of MOSAIC implemented in GEM-MACH does not include the latest ASTEM solver updates (e.g. the bisection method option for the phase state iteration), which are intended to increase execution speed. The time cost of calling the equilibrium HETV routine for each size bin instead of using a bulk formulation is much smaller than for the implemented version of MOSAIC and only doubles the model time-step duration.

There are advantages to using a non-equilibrium approach. Some reduction of model bias relative to observations for sulfate, nitrate and ammonium can be expected and is found by Rosanka et al. (2024). The aerosol pH is impacted as well. There is improved partitioning of nitrate between the fine and coarse modes (Feng and Penner, 2007). A hybrid approach (Capaldo et al., 2000) where dynamical mass transfer is restricted to particulate with diameters greater than 1 µm is substantially cheaper numerically and can approximate the full dynamical approach to a high degree. This scheme did not account for the Kelvin curvature effect, but it can be introduced as a correction (e.g. Zieger et al., 2017).

MOSAIC has a more accurate representation of hydration hysteresis compared to CAM. However, in the regionally aggregated comparisons presented here (Figures 5-9) the impact of dynamical mass transfer, the Kelvin curvature effect and the aerosol water scheme do not result in large differences between the MOSAIC and CAM options in the case of the monthly evolution of ammonium. Missing base cation effects in HETV are more prominent.

In general, model biases compared to observations can reflect other factors such as model resolution (e.g. Mircea et al., 2016), uncertainties in emissions, lack of organic constituent effects, and limitations in cloud process representation including wet scavenging (e.g. Ghahreman et al., 2024).

The purpose of introducing MOSAIC into GEM-MACH is to expand capacity for improved process representation. This includes aerosol heterogeneous chemistry which is sensitive to gas-aerosol mass transfer, acidity and hydration. The improved prediction of aerosol number in the nucleation and Aitken modes facilitates coupling of the aerosol and cloud schemes in the model. Such processes and coupling were not included in the evaluation presented here and this limits the difference compared to CAM. The numerical cost of MOSAIC should not be considered a negative in this context. Improved process representation is numerically expensive which requires machine learning and meta-modeling approaches to mitigate (e.g. Tang and Dobbie, 2011; Gorkowski et al., 2019).

Recent development work has substantially improved HETV (HETP; Miller et al., 2024). HETP is based on ISORROPIA-II (Fountoukis and Nenes, 2007) and includes base cations. This version improves the predicted nitrate and ammonium. HETP is roughly twice as fast as HETV, which justifies development of an improved implementation of CAM. In addition to calling the thermodynamics routine for each size bin, a double-moment formulation should be adopted. This requires introduction of aerosol number prognostic tracers and substitution of the coagulation and sectional adjustment schemes with double-moment variants such as the ones used by MOSAIC. A tradeoff between numerical cost and process accuracy for long term regional scale simulations with an improved CAM in GEM-MACH is justified. This upgraded CAM could see operational use. However, it cannot replace the more accurate mass transfer characteristics of MOSAIC which is intended for policy scenario and research

simulations.

*Code and Data Availability*. The GEM model is free software which can be redistributed and/or modified under the terms of the GNU Lesser General Public License as published by the Free Software Foundation. It is available from https://github.com/ECCC-ASTD-MRD/gem/ (last access September 11, 2024) and branch 5.1 was used. GEM-MACH includes an additional source code tree which is called via an interface routine in GEM. The modified version of this source tree and data used for the analysis presented herein is available online at https://doi.org/10.5281/zenodo.13787463 (Semeniuk, 2024). Due to large data volume and non-standard file format raw model data is not publicly accessible but is available on request from Kirill Semeniuk (kirill.semeniuk@ec.gc.ca).

*Supplement*. There is a supplement for this article.

*Author contributions*. KS implemented MOSAIC in GEM-MACH, designed the numerical experiments, performed all of the simulations, carried out the analysis presented herein and wrote the manuscript. AL contributed to the analysis by suggesting additional model-data comparison metrics and assisted with the station network comparison routines which he developed. AD contributed to the layout and revision of the manuscript.

*Competing interests*. The contact author has declared that none of the authors has any competing interests.

*Acknowledgements*. The authors would like to thank the individuals and organizations responsible for the establishment and maintenance of the AERONET, AMON, AQS, CAPMoN, CASTNET, CSN, IMPROVE and NAPS stations the data from which was used in this study.

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
