# Peer review of "Implementation of the MOSAIC Aerosol Module (v1.0) in the Canadian Air Quality Model GEM-MACH (v3.1)"

_EGUsphere, 2024_

## Author Response (AR1)

**Reply to Reviewer #1**

The authors would like to thank the reviewer for helpful comments. Our replies are highlighted below in blue, reviewer comments are in black Italics.

*It seems that the MOSAIC scheme with the EMR dry deposition produces overall the best set of results, albeit with a computational cost 3x the CAM run. Some improvements to the CAM configuration seem to be possible. A difference in the thermodynamics (no treatment of cations in CAM while they are treated in MOSAIC) are important to the nitrate simulation. There does not seem to be sufficient information given to judge the "best" approach. For example, if the goal is 5 day forecast of PM and it takes 1 hour to produce with CAM and 3 hours to produce with MOSAIC that seems insignificant. If it is 1 day versus 3 days then it matters. I can't judge this given what was in the paper.*

We have revised the Introduction and Conclusion sections (items 1 and 19 in the List of Changes) to clarify the point raised by the reviewer. The purpose of MOSAIC in GEM-MACH is to improve the scenario simulation capability as opposed to operational forecasts. There are numerous missing process representations in GEM-MACH such as organic thermodynamics and heterogeneous chemistry. It is not possible to include these important processes in the operational model due to computational cost and time penalty. However, GEM-MACH is used for policy scenario simulations in a research mode. Here the time penalty is not critical, and the computational cost of better process representation is justified.

*Specific points:*

*Line 70 - what is the implication here of chemical and aerosol tracers not being transported by convection? This seems like it would be important to determining the vertical profile. Can you add anything about why this is not a first order issue to be addressed?*

This is indeed an important limitation of GEM-MACH but it is mitigated to a great degree by resolved transport by synoptic and mesoscale systems (e.g. Polvani and Esler, 2007, doi: 10.1029/2007JD008555; Lyons et al., 1995, doi: 10.1016/1352-2310(94)00217-9). However, ventilation of the atmospheric boundary layer by shallow convection is not properly captured and is a model bias (Polavarapu et al., 2016, doi: 10.5194/acp-16-12005-2016). We have added more discussion of this aspect in the text and included pertinent references (item 3). Transport issues are indicated by the disagreement between the VSD in the model and from AERONET. But given the vertical distribution of aerosols, which decreases rapidly with altitude, these differences are likely due to issues with transport in the planetary boundary layer (items 13 and 15).

*Line 213 - I am confused about what is being discussed here, noting particularly the claim to be using GEOS-5 driving data. Later on line 264 you say you are getting dynamical information from GDPS. Is something in error here?*

The boundary conditions were produced by a MOZART-4 simulation which is a CTM distinct from GEM-MACH. MOZART-4 is driven by GEOS-5 data, GEM-MACH is driven by GDPS and other ECCC produced inputs based on the operational forecast model GEM. We have revised the text to make this distinction clearer (item 6).

*Captions to figures 2b and 2c need to be corrected to refer to figure 2a (not 3a)*

This has been corrected.

*The numerical values in the labeling on Figure 5 are impossible to read even blown up on my screen. I suggest that you tabulate these either in the main text of the supplement. This would also give a more clear presentation of "best" values. Same comment applies to figures 6-9.*

We have increased the font size in the figures (item 10). We have also added section S4 in the Supplement which tabulates the similarity metrics in Figure 5 (Table S4.1) and has more discussion of the differences.

*Line 625 - I find the appeal to non-sphericity to be unconvincing (although it could even be right). Why not consider that you just don't have a great simulation of coarse aerosol amounts? You don't seem to be evaluating it particularly here and so maybe things like dust and sea salt that gets into your domain from remote sources isn't done well (or local dust size distribution).*

We have changed the text to de-emphasize this explanation and point out the underestimation aspect. However, in the process of reprocessing the data in response to a comment by Reviewer #2 we discovered an error in that Level 1.5 instead of 2.0 data was used. The revised Figure 10 has the coarse mode peak at around 6 μm instead of 8 μm which is much closer to the model peak at around 5 μm. We have also added more discussion on the wind-blown dust aspect, but it fails to explain the large low bias in the model accumulation mode (items 13, 14 and 15).

*Line 711 - type - "presented in Sections 4.1 and 4.2"*

This has been revised.

*Figure 14 - State explicitly here that the difference is EMR-REF*

This correction has been added.

**Reply to Reviewer #2**

The authors would like to thank the reviewer for useful comments. Our replies are highlighted in blue and the reviewer comments are in black Italics.

*The paper is very detailed and provides a lot of useful information. However, it lacks a clear message and a prioritisation of the importance of the findings for the reader, who is not directly working with the particular version of the GEM-MACH model.*

*Overall, the paper is too long, and I suggest omitting certain evaluation aspects for the sake of clarity.*

We have attempted to address the clarity aspect raised by the reviewer. The additional content required to answer the reviewer's comments as discussed below has been put in the Supplement to reduce the size of the paper. However, we believe that pruning the existing text and analysis for brevity undermines our article. In particular, the comparison between the REF and EMR runs establishes the need for a new dry deposition scheme for aerosols. Improving one process representation in a model may require doing this for other processes as well.

*On the other hand, the discussion of the reasons for the differences in the results and their attribution to the differences in the modelling approach can be strengthened. I strongly recommend to add a more concise overview (e.g. table) of the key differences and communalities in the aerosol modelling (order of the scheme, hydration, thermodynamic system solution regimes, role of cations etc )*

We have introduced the table (Table 1) suggested by the reviewer in Section 3.1 (item 4 in the List of Changes). But we have retained this content in the text.

*The comparison of the two dry deposition configurations (i.e. four model runs have be compared rather than two) does not add a lot of useful information because the the science of the dry deposition scheme is not sufficiently discussed. I suggest to present only the runs with the EMR update of the Zhang scheme because it seems to lead to better results.*

We have added additional discussion and references on the improvements of the Emerson et al. (2020) scheme over the Zhang et al. (2001) scheme used in the reference GEM-MACH. The Zhang scheme has a high bias in the scavenging of the accumulation and Aitken modes. In fact, it is an outlier in having the scavenging minimum occurring above 1 µm when compared to other schemes, including Petroff and Zhang (2010; doi:10.5194/gmd-3-753-2010) (see Figure 1 in Plem et al., 2022; doi:10.1029/2022MS003050). A detailed discussion of the science of dry deposition is beyond the scope of our paper. The Emerson paper makes sufficient justifications, based on extensive new observational data, for the updated parameters it provides. In this it is supported by Pleim et al. (2022) and Jiang et al. (2023; doi: 10.5194/acp-23-4361-2023). The second paragraph of Section 3.7 has been extended to include these references and highlight the shift in the scavenging minimum (item 7).

Our results also support the Emerson et al. (2020) scheme since the Zhang et al. (2001) scheme is negatively impacting the more accurate MOSAIC size distribution while working much better with the biased CAM size distribution in GEM-MACH. This undermines the performance of MOSAIC

compared to CAM.   We consider an evaluation of the impact of the Zhang scheme to be relevant information that serves as justification for using the Emerson scheme and this can be considered a significant result of our study.

*The comparison of the seasonal maps of surface nitrate, ammonium and sulphate with a model-observation composite product (Donkellaar et al., 2019) should not be called "observations" because it is based on GEOS-Chem, surface observations and uncertain satellite-based PM fields. The usefulness of that composite product for the evaluation should be properly discussed before it used. The observation included in the product are more or less the same observations used by the authors themselves in the next section, which is to some extent a duplication.  However, it is good to compare the simulated  fields as map to show the differences also in areas without in-situ observations (e.g. over oceans)*

We have replaced the term "observations" with "observation product" to refer to the Donkelaar et al. synthesis product in Section 4.1 of the text (item 8).   But the current text already makes it clear that this is not a direct observation product.  There are no direct observational maps of surface aerosol constituents and objective analysis products have value and are used routinely.   The station data is point-wise analysis which we do not consider a duplication.

*The comparison with the size resolved AERONET VSD product offers interesting insight but also the reliability of that data set needs to be further addressed. I also suggest including in the paper a more basic evaluation with AERONET AOD or AE observations. The same holds for a verification with standard PM2.5 observations, which should be reported on.*

We have included an evaluation for measurement uncertainty in Section S3 of the Supplement for surface network stations where possible and refer to these values in the text (item 9).  We have chosen not to revise Figures 5-9 to include these measurement uncertainties.  Networks, aside from AQS, do not supply measurement error data (precision and bias) in the files they provide, or it is missing.  AERONET has uncertainty analysis data which can be downloaded on a station-by-station basis which we have done for the 52 stations used in our paper.  For some reason this uncertainty data is not included in the regular aggregated station data.   The text has been revised to refer to AERONET uncertainty and new Section S5 in the Supplement includes the VSD size distribution uncertainty plotted in Figure S14 (item 12).

We have added content to compare AERONET AOD, single scattering albedo and AE fields with GEM-MACH output in the article (new Section 4.4) and Section S6 in the Supplement (item 17). This analysis is limited by our offline aerosol optical properties code which is based on Bohren and Huffman (1983) and does not consider aerosol mixing state and chemical composition.   However, it serves to underscore the difference in the size distribution between CAM and MOSAIC which leads to better agreement overall with observations for MOSAIC.  As with the VSD data, this is a limited sampling that is more qualitative in nature.

In the course of carrying out this additional analysis we found a mistake in our original VSD processing where we used Level 1.5 instead of 2.0 products.   The error data is only available for level 2.0 and we have updated Figure 10 accordingly.  The coarse mode peak is now at about 6 μm instead of 8 μm which agrees better with the model.

*It would be very welcome if the authors could identify more clearly in the conclusions, which aspects of the MOSAIC module seems to be most important for achieving the reported improvements in model performance.*

We have added a short paragraph at the beginning of the Conclusions section to summarize the discussion present in the rest of the section (item 18).

**List of Manuscript and Supplement Changes**

Pages refer to the revised manuscript and changes are marked in red in the marked up version.

1) Revised first paragraph on page 2 to include: *The revised model is intended for policy scenario and research simulations instead of operational forecasting due to the high numerical cost of these process representation improvements.*

2) Added reference to Fuchs and Sutugin (1971) in the second paragraph on page 2.

3) Added new paragraph and references on page 3: *This is mitigated to a significant degree by vertical transport associated with resolved mesoscale and synoptic systems (e.g. Polvani and Esler, 2007; Lyons et al., 1995).  However, lack of tracer convection does introduce a low bias in the transfer of tracer mass into the free troposphere from the atmospheric boundary layer via shallow convection. This aspect is discussed in the study of Polavarapu et al. (2016) which focuses on CO2 transport in a GEM-MACH derived model.*

4) Added new table comparing CAM and MOSAIC on page 4.

5) Added references to Zdanovskii (1948) and Stokes and Robinson (1966) on page 6.

6) Included new text in the first paragraph on page 8: *MOZART is a chemistry transport model using different chemistry and physics routines than those in GEM-MACH.  In the future, global GEM-MACH simulations will be used to produce high temporal and spatial resolution boundary conditions.*

7) Added new text and references in the first paragraph on page 9: *The shift in the scavenging minimum from sizes above 1 µm to around 0.1 µm in the Emerson scheme results in better agreement with observations, which is supported by other studies (e.g. Pleim et al., 2022; Jiang et al., 2023).*

8) Section 4.1 has references to "observations" replaced by "observation product" in the text and figure captions.

9) The second paragraph on page 21 has been revised to include: *Measurement uncertainties for the observation networks considered here are given in Table S3.1 in Section S3 of the Supplement. In general, the CSN, IMPROVE and NAPS concentrations of sulfate and nitrate have uncertainties around 15% or less. For CAPMoN and CASTNET the sulfate uncertainty bound is 9-13% and for nitrate it is 18-25%.  For ammonium the uncertainty bound is under 15% for networks except for CSN where it is 20%.*

10) The font size in Figures 5-9 has been increased for readability.   The third paragraph on page 21 has been revised to include: *The similarity metrics from Figure 5 are tabulated in Table S4.1 with some additional discussion in Section S4 of the Supplement.*

11) Figures 6, 7, 9 and 10 have been reformatted to portrait from landscape.

12) New paragraph on pages 31-32: *Uncertainty estimates for the AERONET VSD mode parameters (volume median radius and standard deviation) are available (Sinyuk et al., 2020) and have been*

*used to produce size-dependent distribution errors (see Section S5 in the Supplement for details) which are shown in Figure S14. AERONET fine mode size distribution retrievals have been compared with in situ aircraft measurements by Schafer et al. (2019) in several US regions.  Generally, the AERONET deviation in the peak concentration radius is 5.2% and in the VSD fine mode width is 15.8%.   Roger et al. (2022) created a model based on size distribution parameters from 851 AERONET globally distributed stations for use with satellite measurement validation.   The fine and coarse mode particle volume concentrations have an uncertainty under 10% except for 440 nm optical depths less than 0.05.   The uncertainty in the particle volume median radius in the fine and coarse modes is about 0.02 µm and 0.32 µm, respectively.   The standard deviation of the size distribution uncertainty is about 0.04 and 0.044 for the fine and coarse modes, respectively.  For the data considered here these uncertainties correspond to about 10.5% for the fine mode peak radius and 9.2% for the mode width.  For the coarse mode the values are 11.3% and 7%, respectively. These errors are much higher than the provided in the AERONET uncertainty analysis product (see Fig. S14, top left panel).  However, the deviation of the model from observations is much larger than the AERONET VSD uncertainties.*

13) The third paragraph on page 32 has been revised to change the AERONET VSD coarse mode peak diameter from 8 to 6 µm, to remove the word "substantially" and to include: *The EMR results show very little difference compared to the REF case and are not shown... The model substantially underestimates the wet mass in the coarse mode which may account for the peak size bias in this mode.  This could involve wind-blown dust which is not included in our model but a low bias is found in models that do and is related to problems with emissions, turbulent transport and dry deposition (e.g. Adebiyi and Kok, 2020). Coarse mode wind-blown dust tends to maximize in spring and summer (e.g. Hand et al., 2019) which is broadly consistent with the model bias. The sea-salt size distribution is expected not to have such a bias.*

14) The first paragraph on page 33 has been revised to include: *(see Figure S15 in Section S5 of the Supplement) which is reflected in the monthly standard deviation error bars in Figure 10.  The largest deviation of the model from observations is in summer and applies to both the fine and coarse modes.  The PM2.5 fraction of wind-blown dust maximizes in spring (e.g. Park et al., 2010) and accounts for under 5% of the mass so it cannot explain the disagreement.  The model also has less inter-monthly variation of the VSD in each season (Fig. S15)*

15) The second paragraph on page 33 includes the following revision: *In addition, the disagreement between the model and station observations at the surface in terms of total aerosol volume is much smaller than the column VSD difference (not shown).  This implies that there is a model deficiency in the vertical transport of tracers.*

16) The third paragraph on page 33 is revised to include: *uses log-normal distributions for inversion, and this appears to underestimate values in the small size limit of each distribution resulting in a low bias around 0.1 µm and 1.0 µm.*

17) A new Section 4.4 has been added on page 34.  The original Section 4.3 has been renamed Section 4.5 called "Model Size Distribution Analysis".   The new text is:

*4.4 AERONET Aerosol Optical Properties*

*In this section we compare the model column aerosol optical depth (AOD), single scattering albedo (SSA) and Angstrom exponent (AE) with AERONET observations for the stations considered in the previous section. A simple Mie scattering scheme based on the code in Appendix A of Bohren and Huffman (1983) is used for model profiles. This code does not consider composition and mixing state effects and applies to homogenous spherical particles characterized by a volume distribution and water content. The AERONET AOD and SSA uncertainties are discussed in Section S6 of the Supplement.*

*Figure S16 shows the monthly mean AOD and SSA for 440 nm, 675 nm, 870 nm and 1020 nm wavelengths for EMR MOSAIC, EMR CAM and AERONET. Due to the difference in the accumulation mode size distribution, MOSAIC exhibits an increased AOD compared to CAM for all months of the year at 440 nm. For longer wavelengths MOSAIC shows a decrease compared to CAM from May to November. This is consistent with the largest disagreement in the accumulation mode occurring in summer (Fig. S15) when a realistic size distribution minimum around 1 μm fails to form with the CAM option. In the case of SSA, MOSAIC exhibits lower values compared to CAM throughout the year for all wavelengths. The fit to AERONET values is best in summer months but the seasonal cycle seen in the observations is not captured. SSA depends on the composition, mixing state and size distribution of aerosols (e.g., Tian et al., 2023). The excessively high values of SSA (over 0.9) in the model during winter indicate that there is a composition and mixing state difference that cannot be captured by the simple Mie scattering scheme we use. Without a more sophisticated aerosol optical properties model, we cannot assess if there are seasonal biases in the aerosol composition.*

*The column AE averaged over 440-870 nm is shown in Figure S17. MOSAIC has a much better fit to AERONET AE compared to CAM. CAM has values that are too low from April to November. The AE reflects differences in the size distribution and CAM has a poor fine mode distribution compared to MOSIAC and AERONET from spring to fall (Fig. S15).*

18) The first paragraph on page 41 has been revised to include: *MOSAIC improves the size distribution compared to CAM in the accumulation mode. This is due to the use of a double-moment scheme with thermodynamics applied to each size bin. On account of deficiencies in the Zhang et al. (2001) dry deposition scheme this results in a degradation of sulfate compared to CAM in the reference case. This issue is removed using the Emerson et al. (2020) dry deposition parameters. The MOSAIC impact on aerosol composition derives from the inclusion of base cations which are not present in HETV. We expand on these aspects below.*

19) A new paragraph five has been added on page 42: *The purpose of introducing MOSAIC into GEM-MACH is to expand capacity for improved process representation. This includes aerosol heterogeneous chemistry which is sensitive to gas-aerosol mass transfer, acidity and hydration. The improved prediction of aerosol number in the nucleation and Aitken modes facilitates coupling of the aerosol and cloud schemes in the model. Such processes and coupling were not included in the evaluation presented here and this limits the difference compared to CAM. The numerical cost of MOSAIC should not be considered a negative in this context. Improved process representation*

is numerically expensive which requires machine learning and meta-modeling approaches to mitigate (e.g. Tang and Dobbie, 2011; Gorkowski et al., 2019).

20) A new Acknowledgements section has been added: *The authors would like to thank the individuals and organizations responsible for the establishment and maintenance of the AERONET, AMON, AQS, CAPMoN, CASTNET, CSN, IMPROVE and NAPS stations the data from which was used in this study.*

21) Section S2 of the Supplement has been revised to point to measurement uncertainty: *The measurement uncertainty for these networks is given in Table S3.2 in Section S3 based on acceptance criteria and precision and bias data for AQS, NAPS and AMON. Based on AQS the measurement uncertainty is under 10% for O3 and SO2. For NO2 it is under 15% which falls within the acceptance bound for NAPS. However, AQS and NAPS NO2 measurements are subject to NOz interference (Lamsal et al., 2008). The impact is small where NOx dominates NOy but can be substantial away from NO sources and dependent on local concentrations of HNO3 and other NOy species. For HNO3 the measurement uncertainty is likely under 15% based on the values provided by McNaughton and Vet (1996) for CAPMoN. CASTNET uses a similar filter pack system. The NH3 uncertainty falls in the 10-20% range as well.*

22) New Sections S3, S4, S5 and S6 as well as new Figures S14, S15, S16 and S17 have been added to the Supplement.

23) Instances of "two-moment" have been replaced by "double-moment" for consistency with the language in "single-moment".

24) Instances of "standard error" have been replaced by "standard deviation" in figure captions. Standard error was not calculated for the error bars in the relevant figures.

25) References with "e.g.," prefixes have been revised to remove the comma.

---

## Author Response (AR2)

**Reply to Reviewers**

There were no posted comments by the reviewers to the revised manuscript submitted on March 25, 2025. Therefore, the authors have not made any revisions and any replies to the reviewers. However, sub-figure references for figures 1, 2 and 3 (fig. 1a, b, c, etc.) have been changed to figures 1 through 8 and the other figure numbers and references modified accordingly.